# The Physiological and Pathological Role of Acyl-CoA Oxidation

**DOI:** 10.3390/ijms241914857

**Published:** 2023-10-03

**Authors:** Sylwia Szrok-Jurga, Aleksandra Czumaj, Jacek Turyn, Areta Hebanowska, Julian Swierczynski, Tomasz Sledzinski, Ewa Stelmanska

**Affiliations:** 1Department of Biochemistry, Faculty of Medicine, Medical University of Gdansk, 80-211 Gdansk, Poland; szrok@gumed.edu.pl (S.S.-J.); jacek.turyn@gumed.edu.pl (J.T.); areta.hebanowska@gumed.edu.pl (A.H.); 2Department of Pharmaceutical Biochemistry, Faculty of Pharmacy, Medical University of Gdansk, 80-211 Gdansk, Poland; aleksandra.czumaj@gumed.edu.pl; 3Institue of Nursing and Medical Rescue, State University of Applied Sciences in Koszalin, 75-582 Koszalin, Poland; juls@gumed.edu.pl

**Keywords:** beta-oxidation, peroxisomal fatty acid oxidation, acyl-CoA, fatty acid metabolism

## Abstract

Fatty acid metabolism, including β-oxidation (βOX), plays an important role in human physiology and pathology. βOX is an essential process in the energy metabolism of most human cells. Moreover, βOX is also the source of acetyl-CoA, the substrate for (a) ketone bodies synthesis, (b) cholesterol synthesis, (c) phase II detoxication, (d) protein acetylation, and (d) the synthesis of many other compounds, including N-acetylglutamate—an important regulator of urea synthesis. This review describes the current knowledge on the importance of the mitochondrial and peroxisomal βOX in various organs, including the liver, heart, kidney, lung, gastrointestinal tract, peripheral white blood cells, and other cells. In addition, the diseases associated with a disturbance of fatty acid oxidation (FAO) in the liver, heart, kidney, lung, alimentary tract, and other organs or cells are presented. Special attention was paid to abnormalities of FAO in cancer cells and the diseases caused by mutations in gene-encoding enzymes involved in FAO. Finally, issues related to α- and ω- fatty acid oxidation are discussed.

## 1. Introduction

Fatty acids (FAs) are critical compounds for the health control and development of the human body due to their participation in cellular metabolism, especially energy production (ATP synthesis), metabolism regulation, and cell proliferation. They are (a) building blocks for complex lipids in cellular membranes, (b) precursors for signaling molecules, such as eicosanoids, (c) allosteric regulators of metabolic pathways, (d) substrates for protein acylation, and (e) ligands for transcription factors. FAs are also responsible for lipotoxicity and contribute to the release of proinflammatory molecules, which play an important role in many diseases. Moreover, an increase in citrate, isocitrate, and malate production associated with free fatty acid (FFA) β-oxidation (βOX) leads to increased NADPH levels in some cells. Cytosolic isocitrate dehydrogenase (which catalyzes the conversion of isocitrate in the presence of NADP to α-ketoglutarate and NADPH) and a cytosolic malic enzyme (ME) (which catalyzes the conversion of malate in the presence of NADP to pyruvate and NADPH) play an important role in NADPH homeostasis.

The most important sources of FAs found in humans include dietary supply, mainly triacylglycerols, and de novo synthesis, mainly from glucose [1].

As already mentioned, FAs serve a predominant role as substrates for ATP production in many human and animal organs, including the heart, skeletal muscle, kidney, and liver. Over 20 proteins are involved in the uptake, activation, transport into the organelles (mainly mitochondria and peroxisomes), and finally, fatty acid oxidation (FAO). The most important process of FAO-βOX occurs primarily in the mitochondria of many organs and, to a lesser extent, in peroxisomes, mainly in the liver and kidney. In peroxisomes, not only βOX but also α-oxidation takes place. Alfa oxidation produces a fatty acyl CoA, one carbon shorter [2]. From a practical point of view, this process plays an important role in the oxidation of phytanic acid (a compound present in the human diet, originating mainly from ruminant animals and fish) [3]. ω-oxidation undergoes in microsomes (smooth endoplasmic reticulum) [4]. In this process, FAs are degraded starting from the end methyl group (so-called ω-carbon) of FAs, and the CYP (cytochrome P-450) family is involved. ω-oxidation is considered a rescue process for some genetic diseases in humans, in which mitochondrial and peroxisomal FA oxidation is impaired. Interestingly, phytanic acid also undergoes ω-oxidation [2].

The energy production from FAs is strictly associated with the mitochondrial βOX. The intensity of βOX is controlled by a plethora of regulatory factors, including the supply of nutrients and the action of several hormones, including insulin, glucagon, catecholamines, triiodothyronine, and cortisol. The crucial regulator of FAO is peroxisome proliferator-activated receptor α (PPARα) [5]. PPARα is a transcription factor that functions as a heterodimer in complex with the retinoid X receptor α (RXRα) and binds via the PPARα DNA-binding domain (DBD) to the PPRE (peroxisome proliferator response element) sequence in the promoter region of target genes involved mainly in hepatic and cardiac muscle FA and FAO [6]. The initiation of transcription by PPARα (similar to other PPARs) requires its activation. Briefly, in its inactive form, the PPARα-RXRα complex is associated with corepressors [7]. The complex activation occurs following ligand binding [8]. A wide range of lipophilic molecules can activate PPARα. These include natural saturated, unsaturated, and polyunsaturated fatty acids (PUFAs) and synthetic ligands, collectively called PPARα activators [7,9]. The natural ligands show different binding affinities and strengths of PPARα activation. The potent PPARα ligands are unsaturated fatty acids, including omega-3 eicosapentaenoic acid (20:5, ω3), docosahexaenoic acid (22:6, ω3), and phytanic acid [10,11]. The natural and synthetic ligands (pharmacological ligands, for instance, fibrates) directly bind to PPARα via the ligand-binding domain (LBD). The ligand binding to a nuclear receptor causes the release of corepressors and begins the recruitment of coactivator complexes to the PPARα-RXRα, which enables the activation of the expression of genes involved in FAO [7]. PPAR*α* is expressed at the highest level in hepatocytes, cardiomyocytes, enterocytes, and kidney proximal tubule cells, which are involved in the increased FAO [12], as we describe in this review. Other members of the PPARs family—PPARβ/δ and PPARγ—are involved in the regulation of different processes generally associated with lipid metabolism. PPARβ/δ participates in the activation of FAO [13]. It has been observed that expression of the *PPARβ/δ* genes increases in skeletal muscles after fasting and endurance exercises, which promotes the transition from glucose, as the primary source of energy substrate, to lipids [14,15,16,17]. In comparison, PPARγ plays an important role in adipogenesis, lipid uptake, triacylglycerols (TAG) storage, and lipid droplet formation [18].

In this review, we first described a general aspect of the FAs transport (into the cells and then mitochondria) and activation. Then, we concentrated on FAO under physiological and pathological conditions in the liver, heart, skeletal muscle, kidney, and other organs. Special attention has been paid to FAO abnormalities in cancer cells and the diseases caused by mutations in genes encoding enzymes involved in FAO.

### 1.1. Uptake and Activation of Fatty Acids

In blood, FAs are present as components of lipids in (a) cell membranes (mainly in erythrocytes and white blood cells), (b) lipoproteins (mainly in chylomicrons, VLDL, LDL, and HDL), and (c) FFAs mostly bound to albumin. The major FAs in the whole lipids in the blood are palmitic acid (C16:0), stearic acid (C18:0), oleic acid (C18:1), linoleic acid (C18:2), and arachidonic acid (C20:4) [19,20]. The concentration of FFAs in the serum increases during exercise or fasting, and they are mainly used as FAO substrates in skeletal muscles, the heart, liver, and kidney [21]. The physiological FFA concentration in blood is around 0.2–0.5 mmol/L [22]. Due to their low solubility in H_2_O (1–10 nmol/L, depending on FA chain length), FFAs (mainly long-chain—LCFAs and medium-chain—MCFAs) are attached to the albumin [23]. Binding the FFAs to the albumin (a) enables transport in the blood and (b) protects human organs against some pathologies, including insulin resistance, non-alcoholic fatty liver disease, atherosclerosis, and heart dysfunction [24,25]. FFAs are translocated from the albumin FFA complex into the target cell (cells where FFAs are metabolized) cytosol across the endothelial layer of the blood vessels [26]. In the liver, the sinusoidal endothelial cells are fenestrated and do not have a basement membrane, so the absorption of FFAs is much easier than in other organs [27]. The transfer of FFAs from the blood to other cells, for instance, cardiomyocytes, seems to be more complicated since the endothelial wall in the heart capillaries is not-fenestrated and the FFAs are transferred through three lipid membranes: two endothelial (in and out of the endothelial cells) and one myocyte membrane (transported into the cell). Arts et al. proposed a model of FFA translocation across heart capillaries into cardiomyocytes, where FFAs bind to compartment-specific carrier proteins [28]. According to this model, the crossing of the plasma membrane remains under the control of several proteins, including (a) cluster of differentiation-36 (also known as FA translocase—CD36), (b) FA-binding protein—FABPm, and (c) FA-transporting protein—FATP (Figure 1). These proteins enable the cell to control the inflow of FFAs precisely. They increase the uptake of FFAs at the beginning of muscle contraction, even if the concentration of FFAs in the blood is low. Moreover, they also prevent the entrance of excess FFAs into the cell and help to select FFAs according to the cell’s demand. It should be noted that FFAs may also translocate into the target cell by a flip-flop system driven by the FFA concentration gradient [28].

The delivery of FFAs to the cells and their activation before usage in several cellular processes involves many proteins, including enzymes (Figure 1). Among these proteins are FATPs, which exhibit acyl-CoA synthetase activity. These two functions of FATPs (transport and activation) enable the immediate utilization of FFAs in the cell. Other proteins, like FABPm, CD36, and a family of acyl-CoA synthetases (ACSs), form an integrated system of the transport and activation of LCFAs, MCFAs, and short-chain FAs (SCFAs). FABPc proteins are also involved in binding FFAs in the cytosol (Figure 1).

FFAs are activated by specific ACSs. After activation, some FFAs become bound by the acyl-CoA-binding proteins (ACBPs). The binding of FFAs is responsible for channeling acyl-CoA to particular cellular compartments and processes. Acyl-CoA’s minor pool is deacylated by acyl-CoA diesterases (ACOTs) [29,30]. However, the physiological significance of deacylation is unknown. Very recently, it has been reported that ACOT1 knock-out partially protects mice from high-fat diet-induced weight gain by increasing energy expenditure [31]. Thus, these results suggest that inhibition of ACOT1 could prevent obesity during caloric excess.

**Figure 1 ijms-24-14857-f001:**
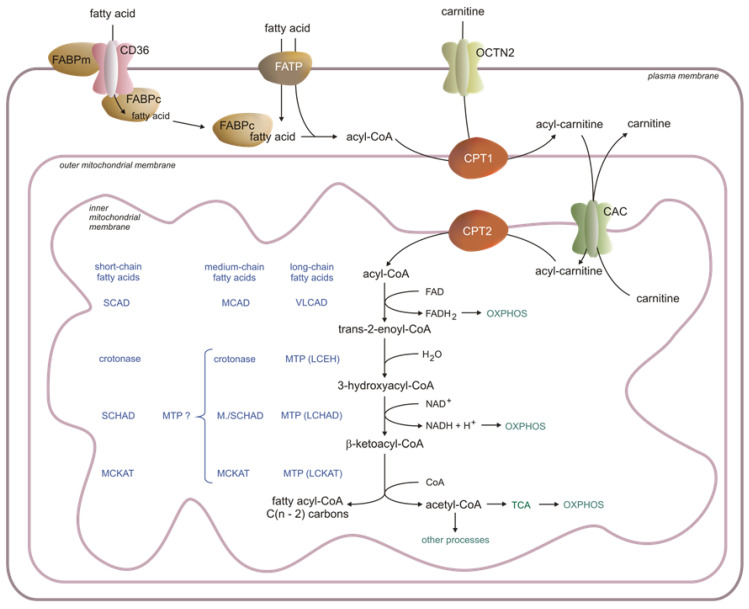
Fatty acid transport and metabolism in the cell. CAC—acylcarnitine translocase, CP—carnitine palmitoyltransferase, FABP—fatty acid-binding protein, LCEH—long-chain enoyl-CoA hydratase, LCHAD—long-chain fatty acid hydroxy acyl-CoA dehydrogenase, LCKAT—long-chain fatty acid β-ketothiolase, MCAD—medium-chain acyl-CoA dehydrogenase, MCKAT—medium-chain ketoacyl-CoA thiolase, OXPHOS—oxidative phosphorylation, SCAD—short-chain acyl-CoA dehydrogenase, SCHAD—short-chain hydroxy acyl-CoA dehydrogenase, TCA—Krebs cycle, VLCAD—very-long-chain acyl-CoA dehydrogenase, OCTN2—carnitine transporter, present in the heart, skeletal muscle, and kidney (hepatocytes have a different translocator with low affinity and high capacity), FABPm—membrane fatty acid-binding protein, FABPc—cytosolic fatty acid-binding protein, MTP—mitochondrial trifunctional protein, MTP ? – possible involvement of MTP protein, CD-36—fatty acid translocase, FATP—fatty acid transporting protein (the acyl-carnitines are transported across the outer mitochondrial membrane via a voltage-dependent anion channel (VDAC) [32]).

According to the chain length, influencing the hydrophobicity and water solubility of FFAs, four ACS families have been established: (a) short-chain acyl-CoA synthetases (ACSSs), (b) medium-chain acyl-CoA synthetases (ACSM), (c) long-chain acyl-CoA synthetases (ACSLs), and (d) very-long-chain acyl-CoA synthetases (ACSVLs) [33]. An overview of the characteristics of ACS isoforms is presented in Table 1.

Except for lauric acid, MCFAs are activated and oxidized in mitochondria [45,48].

### 1.2. Carnitine Shuttle

#### 1.2.1. Carnitine Palmitoyltransferase 1 (CPT1)

The inner mitochondrial membrane is impermeable to the long-chain acyl-CoAs. Thus, the acyl-CoAs are converted to acylcarnitine in the reaction catalyzed by carnitine palmitoyltransferase 1 (CPT1):acyl-CoA + carnitine → acylcarnitine + CoASH

CPT1 is a hexamer, a part of a protein complex formed and attached to the outer mitochondrial membrane. Other elements of that complex are ACSL and VDAC (voltage-dependent anion channel) [49,50]. Three isoforms of CPT1 are known: CPT1A, CPT1B, and CPT1C. CPT1A is the main CPT1 in the liver, but it is also present in minor amounts in the heart, skeletal muscles, brain, kidneys, lungs, spleen, intestine, pancreas, ovaries, and fibroblasts. It is involved in transporting LCFAs and medium-chain lauric acid (C:12) into mitochondria, though its highest activity is in lauric acid. CPT1B is the dominating form in the skeletal muscles, heart, and testes, and like CPT1A, it is an enzyme transporting LCFAs to mitochondria, with the highest activity in C12-C16 FFAs. CPT1C is a neural form attached to endoplasmic reticulum (ER) membranes. Potentially, it is involved in the neuronal control of thermogenesis in brown adipose tissue (BAT) [51,52]. CPT1C activity is significantly (20–300 times) lower than CPT1A [53,54,55]. CPT1A and B share 62% similarity in the amino acid sequence. Both isoforms differ significantly in activity and regulation [56].

A high-fat diet induces the expression of the *CPT1* gene by the PPARα transcription factors in the liver and muscles [5,57,58]. Insulin, glucagon, and triiodothyronine regulate CPT1 activity in the liver, and the physiological status significantly influences that regulation [57,59,60,61]. The major regulator of CPT1 is malonyl-CoA, a negative allosteric effector of this enzyme. The intracellular level of malonyl-CoA depends on acetyl-CoA carboxylase (ACC—enzyme-synthesizing malonyl-CoA) activity and malonyl-CoA decarboxylase (MCD—enzyme-degrading malonyl-CoA) activity [62,63,64,65]. Malonyl-CoA, an intermediate in palmitate synthesis, inhibits FAO during intensive FFA synthesis. It protects the cell from the immediate oxidation of the newly synthesized FFAs [52]. At a negative energy balance, when the activity of MCD is elevated, CPT1 restores its activity, leading to efficient acylcarnitine synthesis. It should be noted that CPT1B is activated mainly by exercise and is more sensitive to changes in the malonyl-CoA level.

Both LCFAs and MCFAs stimulate CPT1 activity during the exercises [66]. A high-fat diet or fasting induces the expression of the *CPT1* gene by the two independent systems involving PPARα transcription factors or the PGC1α/PPARγ complex in the liver and muscles. The binding site in the *Cpt1* gene for PPARα in the rat liver is located in the second intron and PGC1α/PPARγ in the first intron [5,57,66]. Mutations in PPRE totally eliminate the induction of *Cpt1* gene expression by both regulatory systems [5].

Carnitine is transported from the blood to the cells by the high-affinity OCTN2 carnitine transporter in the cell membrane of the heart, skeletal muscle, and kidney (Figure 1) [67]. It should be noted that different types of carnitine transporters with low affinity and high capacity are present in hepatocytes.

#### 1.2.2. Carnitine Palmitoyltransferase 2 (CPT2) and Acylcarnitine Translocase CAC (SLC25A20)

CAC (SLC25A20) transfers acylcarnitines across the inner mitochondrial membrane [68]. CAC forms a functional complex with carnitine palmitoyltransferase 2 (CPT2) in the inner mitochondrial membrane (Figure 1), leading to the transesterification of acyl groups from acylcarnitines to mitochondrial CoAs according to the reaction:acylcarnitine + CoA-SH → acyl-CoA + carnitine

A high acylcarnitine concentration in the intermembrane space drives its translocation into the matrix [68]. The overall role of CPT1, CAC, and CPT2 in the transport of acyl-CoA into the mitochondrial matrix is presented in Figure 1. NO, H_2_S, nonenzymatic acetylations, β-lactam antibiotics, omeprazole (proton pump inhibitor), and heavy metals inhibit CAC [69,70,71,72,73,74,75,76]. PPARα and other transcription factors or transcriptional coactivators (estrogen receptors, PGC1α) activate the transcription of CAC, and polyphenols (antioxidants) increase the effectiveness of βOX. Statins, drugs lowering serum cholesterol concentration, and retinoic acid also increase CAC activity [77,78,79,80].

### 1.3. Mitochondrial β-Oxidation

A few years ago, the mitochondrial βOX was described by Hounten et al. in an excellent review [81]. Briefly, the first step of each βOX round is catalyzed by an acyl-CoA dehydrogenase (AD), producing trans-2-enoyl-CoA. In the next step, the hydration of a double-bond is catalyzed by enoyl-CoA-hydratase (ECH), and the following dehydrogenation by hydroxy-acyl-CoA dehydrogenase (HAD) leads to the production of 3-keto-acyl-CoA. The last step of the cycle is thiolysis. In each round of βOX, one FAD and NAD^+^ accept two electrons each and change into FADH_2_ and NADH, respectively. The electrons are then transferred to the mitochondrial respiratory chain, where oxidative phosphorylation (OXPHOS) occurs. The acetyl-CoA formed may enter the Krebs cycle (TCA) (mainly in the heart, kidney, and skeletal muscle) and other processes (for instance, ketogenesis in the liver) (Figure 1) [82,83]. The acyl-CoAs, which are shorter by two carbons compared to the initial substrate, enter the next round of βOX. The odd-chain FFAs (present in a small amount in human tissue) are degraded, like the even-chain acyl-CoAs, to several acetyl-CoAs (depending on FFAs). However, propionyl-CoA arises from the methyl end of the odd-chain acyl-CoA. Propionyl-CoA is converted via methylmalonyl-CoA to succinyl-CoA, metabolized in the TCA, or converted to glucose in the liver. The amount of propionyl-CoA formed from odd-chain FFAs is very small because the number of such FAs in the diet is relatively low.

Five ADs found in human cells are involved in the first step of βOX. Characteristics of ADs are presented in Table 2.

#### 1.3.1. Oxidation of Long-Chain Acyl-CoA

Oxidation of long-chain acyl-CoA is catalyzed by one of three ADs: a) very-long-chain acyl-CoA dehydrogenase (VLCAD), b) acyl-CoA dehydrogenase DH-9 (ACAD9), and c) long-chain acyl-CoA dehydrogenase (LCAD). VLCAD oxidizes most LCFAs entering mitochondria. This enzyme, bound to the inner mitochondrial membrane, oxidizes C14:0–C22:0 acyl-CoA, although the preferred substrate is palmitoyl-CoA. The presence of an unsaturated bond in FFAs decreases the efficiency of the reaction catalyzed by this enzyme. PPARα is the most important VLCAD regulator, increasing its gene expression. Sirtuins (especially sirtuin 3) may also activate VLCAD through deacetylation [84,85,86,87,88].

ACAD9 is homologous to VLCAD and uses mostly unsaturated long-chain acyl-CoAs as substrates. It is abundant in the brain and liver. Despite the homology of this enzyme with VLCAD, neither enzyme can compensate for each other in their deficiency [85,89]. LCAD is localized in the mitochondrial matrix. It is mainly present in lung alveolar cells. LCAD knockout caused pulmonary surfactant (complex substances, mainly lipids, which play important functions in the alveoli and small airways) dysfunction and increased susceptibility to lung infections [86]. An in vitro investigation showed that some unsaturated and branched-chain acyl-CoA are the principal substrates for LCAD. This enzyme is exceptional among ADs because it tends to leak electrons, producing H_2_O_2_. Its function in organs other than the lungs has not been estimated [90].

Each AD uses FAD as an electron acceptor. Formed FADH_2_ has to be re-oxidized, so the electrons are translocated to a flavoprotein, electron-transferring flavoprotein (ETF), and then ETF-dehydrogenase transfers them into coenzyme Q (CoQ) in the OXPHOS system (Figure 1) [91,92].

The mitochondrial trifunctional protein (MTP) complex participates in the second step of LCFA oxidation. The MTP catalyzes three different reactions in a row. The MTP enzymatic activities are long-chain enoyl-CoA hydratase (LCEH), long-chain hydroxy acyl-CoA dehydrogenase (LCHAD), and long-chain β-ketothiolase (LCKAT). The MTP complex contains “a” and “b” subunits, forming an octamer bound to the surface of the inner mitochondrial membrane due to a strong interaction with membrane phospholipids [93,94]. Subunit “a” contains the enzymatic activities of hydratase and dehydrogenase, whereas subunit “b” contains thiolase activity. This enzymatic complex binds the enoyl-CoAs containing 6–16 carbons, but in the liver, its activity is the highest for C10 and longer acyl-CoAs. The final product of MTP activity is acetyl-CoA and acyl-CoA, which is shortened by two carbons and enters the next cycle of βOX [95].

#### 1.3.2. Oxidation of Monounsaturated and Polyunsaturated Long-Chain Acyl-CoA

Oxidation of monounsaturated long-chain acyl-CoA requires an additional enzyme called 3,2-trans-enoyl-CoA isomerase (ECI), which catalyzes the following reaction:trans-3-enoyl-CoA → trans-2-enoyl-CoA

ECI exists in two isoforms: ECI1 and ECI2. ECI1 is found in mitochondria only, whereas ECI2 is present in mitochondria and peroxisomes. ECI2 has a much higher affinity for LCFAs [96,97,98]. The studies on ECI isoforms were performed using enzymes isolated from rat liver [96] and the ECI1 knock-out mice model [97], and structural studies using X-ray scattering were performed for a human ECI2 isoform [98].

The βOX of polyunsaturated FAs requires a) ECI and b) 2,4-dienoyl-CoA reductase, which catalyzes the following reaction:trans-2,cis-4-dienoyl-CoA + NADPH + H^+^ → tans-3-enoyl-CoA + NADP^+^

Formed tans-3-enoyl-CoA by 2,4-dienoyl-CoA reductase is converted to trans-2-enoyl-CoA by ECI, as presented above.

#### 1.3.3. Oxidation of Medium-Chain Fatty Acids

In the first cycle of MCFA mitochondrial FAO, medium-chain acyl-CoA dehydrogenase (MCAD) catalyzes the initial step. It is a flavoprotein cooperating with ETF and ETF-dehydrogenase. MCAD is a homotetrameric protein localized in the mitochondrial matrix. It is abundant in the human heart, skeletal muscles, and liver [99,100]. The enzymes responsible for the subsequent reactions are not well-defined in humans. It is possible that human MTP participates in the oxidation of medium-chain enoyl-CoAs. However, it is not excluded that MCFAs, which translocate from the cytosol to mitochondria, might be activated and elongated, finally becoming the substrate for MTP [101].

#### 1.3.4. Oxidation of Short-Chain Fatty Acids

The first step of SCFA degradation is catalyzed by short-chain acyl-CoA dehydrogenase (SCAD), a flavoprotein cooperating with ETF/ETF-dehydrogenase. Butyryl-CoA, formed from butyrate produced by gut microbiota, is the major substrate for SCAD, and the product is crotonyl-CoA [102]. SCAD is abundant in the liver, heart, and skeletal muscles. It is a matrix-localized homotetramer. In the liver and kidneys, SCAD also displays oxidase activity, but the significance of this feature is unresolved [103,104]. The other enzymes involved in short-chain acyl-CoA oxidation are crotonase (enoyl-CoA hydratase), medium-chain hydroxy acyl-CoA dehydrogenase, short-chain hydroxy acyl-CoA dehydrogenase (SCHAD), and medium-chain ketoacyl-CoA thiolase (MCKAT), and all those activities are localized in the mitochondrial matrix. Human crotonase uses crotonyl-CoA as a substrate. It is also involved in the metabolism of some amino acids. Crotonase is present in significant amounts in the liver, less in muscles and fibroblasts, and even less in the kidneys and spleen [105,106]. Hydroxyacyl-CoA dehydrogenase is a homodimer localized in the matrix, which produces acetoacetyl-CoA and NADH. The highest activity of this enzyme is present in the heart, muscles, liver, and pancreas [107]. MCKAT catalyzes the last step of short-chain FAO. The activity of MCKATs is present in the mitochondrial matrix, peroxisomes, and cytosol. MCKATs that are present in the matrix of human mitochondria have two main substrates: methyl-acetyl-CoA (metabolized into propionyl-CoA) and acetoacetyl-CoA (metabolized into two molecules of acetyl-CoA).

### 1.4. Peroxisomal FAO

In the liver, FAO takes place both in mitochondria and peroxisomes. However, under physiological conditions, peroxisomal FAO accounts for approx. 5% of total FAO in the liver [108]. Peroxisomal βOX differs significantly from mitochondrial βOX [109,110]. In mitochondria, acyl-CoA dehydrogenases transfer the electrons to ETF, which are subsequently transferred to the mitochondrial respiratory chain and reduce oxygen to water, producing energy (ATP) [82]. In contrast, peroxisome acyl-CoA oxidase 1 (ACOX1) reduces FAD, and electrons are transported directly from FADH_2_ to molecular oxygen, generating hydrogen peroxide (H_2_O_2_) [110]. CoA esters of straight-chain FAs (VLCFAs, LCFAs, PUFAs, and dicarboxylic acids) are preferred substrates for ACOX1, whereas ACOX2 is responsible for the oxidation of branched-chain FAs (BCFA) and the transformation of bile acid intermediates [111]. In addition, Ferdinandusse et al. identified a novel ACOX isoform, ACOX3, which is involved, similar to ACOX2, in the degradation of BCFAs [112].

The oxidation of LCFAs in peroxisomes stops at the level of MCFA-CoAs [110]. MCFA-CoAs can be hydrolyzed to FFAs by the peroxisomal thioesterases. Then, MCFAs, via the pore-forming proteins, leave the peroxisome and are transported to the mitochondria, where βOX is completed. The second way of MCFA oxidation uses carnitine and carnitine acyltransferase with specificity for short- and medium-chain acyl-CoA. Formed acylcarnitines are transported into mitochondria via the mitochondrial CAC [113]. It should be emphasized that peroxisomal FAO needs the participation of mitochondria not only for the oxidation of acetyl-CoA (formed from MCFA-CoAs) but also for the oxidation of NADH [110,114]. For a summary of mitochondrial and peroxisomal βOX, see Table 3.

It has been shown that during peroxisomal βOX (both dicarboxylic and monocarboxylic acids), free acetate is formed, which is preferentially exported from the hepatocyte and used as an energy substrate in other organs [113]. It has been postulated that acetate is formed from acetyl-CoA in a reaction catalyzed by acetyl-CoA hydrolase [92].

#### 1.4.1. Peroxisomal α-Oxidation—Role in Phytol and Phytanic Acid Metabolism

The average Western diet contains approx. (a) 50–100 mg per day of phytanic acid, (b) 10–30 mg per day of pristanic acid, and (c) 10 mg per day of phytol [119]. Phytol mostly comes from nuts [120]. Phytanic acid and pristanic acid are derived primarily from lipids found in beef, dairy products, and fish. [119]. The phytanic acid present in the diet is derived mainly from phytol [121]. Phytol is widely distributed as a constituent of chlorophyll present in the green leaves of plants and trees [3]. Bacteria present in the rumen of ruminant animals cleave the phytol from the porphyrin ring of chlorophyll (the human alimentary tract cannot do this). The released phytol can be oxidized to phytanic acid in the ruminants [3]. Thus, it is clear that phytanic acid is present in meat and dairy products from grass-fed cattle or other ruminants. Phytanic acid can also be derived from vegetables (as phytol bound to chlorophyll) [122]. Moreover, phytyl FA esters are also present in the leaves of some plants, fruits, and vegetables. These compounds are hydrolyzed in the human gastrointestinal tract, providing phytol [123].

Subjects consuming products rich in phytol and phytanic acid oxidize these compounds via α-oxidation because BCFAs containing a methyl group in the 3-position (like phytanic acid) are not metabolized by βOX. First, phytol is oxidized to phytenal in the reaction catalyzed by alcohol dehydrogenase. Formed phytenal is oxidized by aldehyde dehydrogenase to phytenic acid, which in turn is converted to phytenoyl-CoA by acyl-CoA synthetase. In the reaction catalyzed by enoyl-CoA reductase, phytenoyl-CoA is converted to phytanoyl-CoA. Phytanoyl-CoA can also be formed from phytanic acid in the reaction catalyzed by acyl-CoA synthetase. Formed phytanoyl-CoA undergo α-oxidation to 2-hydroxyphytanoylo-CoA, catalyzed by phytanoyl-CoA 2-hydroxylase. This process requires 2-oxoglutarate and Fe^2+^, and O_2_. 2-hydroxyphytanoilo-CoA is converted with the participation of hydroxy acyl-CoA and aldehyde dehydrogenase to pristanic acid, which is activated to pristanoyl-CoA by acyl-CoA synthetase. Next, pristanoyl-CoA undergoes peroxisomal βOX to 4,8-dimethyl nonaoyl-CoA, which in turn is metabolized in mitochondria (Figure 2) [123].

Deficiency of the phytanoyl-CoA 2-hydroxylase impairs the conversion of phytanic acid to pristanic acid (2-methyl BCFAs) and leads to Refsum disease (type IV motor and sensory neuropathy) [124,125]. The only therapy available for that disorder is a diet low in phytanic acid.

#### 1.4.2. Peroxisomes Are Essential for the Degradation of Dicarboxylic Acid Formed during ω-Oxidation in Microsomes

VLCFAs are also oxidized in microsomes via ω-oxidation. In humans, the first step of ω-oxidation is catalyzed by CYP (CYP4F2 or CYP4F3B). Omega-hydroxy-VLCFAs, formed by CYP4F2 or CYP4F3B, can be oxidized to ω-HOOC-VLCFA (dicarboxylic-VLCFA) by alcohol dehydrogenase and subsequently by aldehyde dehydrogenase. Formed HOOC-VLCFA is then oxidized by βOX in peroxisomes. Importantly, the βOX of HOOC-VLCFA is not affected in X-ALD (X-linked adrenoleukodystrophy) patients [2]. Thus, it has been suggested that the peroxisomal βOX of dicarboxylic-VLCFA (formed during ω-oxidation) can provide an alternative route of VLCFA oxidation in X-ALD patients (Figure 3) [2].

#### 1.4.3. Peroxisomal FAO—Potential Role in the Utilization of Toxic FFAs

Peroxisomal βOX is necessary for the oxidation of VLCFAs (≥22 carbons), both saturated and mono- and polyunsaturated [110,113]. These FFAs need to be degraded not because of their role in providing energy but due to the toxic effect of their excessive accumulation (for instance, monounsaturated erucic acid C22:1, present in commonly used canola oil) [113,126]. The βOX of VLCFAs, notably C26:0 and longer-chain FFAs, occurs exclusively in peroxisomes [113].

Abnormalities in the biogenesis of peroxisomes are the cause of Zellweger syndrome. This rare familial disease is characterized by muscle weakness, hepatomegaly, and brain and kidney dysfunction. Goldfischer et al. reported that peroxisomes are absent in the liver and kidney of patients with this syndrome [127]. Consequently, significant amounts of VLCFAs and bile acid synthesis intermediates are accumulated in plasma [125,127,128,129].

Subfamily D of ABC transporters (ATP-binding cassette transporters) in mammals comprises four distinct proteins, namely ABCD1 (adrenoleukodystrophy protein), ABCD2 (adrenoleukodystrophy-related protein), ABCD3 (70 kDa peroxisomal membrane protein), and ABCD4 (peroxisomal membrane protein 69). Three of these, ABCD1-3, are localized solely in peroxisomes and mediate the uptake of substrates into the peroxisome for βOX [115].

ABCD1 and ABCD2 facilitate the transport of VLCFAs or their CoA derivatives into peroxisomes. Interestingly, ABCD1 has a higher specificity for saturated VLCFA-CoA. In contrast, ABCD2 prefers to transport PUFAs, such as C22:6-CoA and C24:6-CoA [130]. However, it is worth adding that the main substrate for ABCD2 in humans is still not completely defined [131]. The ABCD3 transporter is important in transporting branched chain acyl-CoA and bile acid intermediates, e.g., di- and tri-hydroxycholestanoyl-CoA (DHCA and THCA) [132]. *Abcd* genes are under complex regulation at the transcriptional level. The transcription of *Abcd1*, *Abcd2,* and *Abcd3* genes is regulated by PPARα [133,134]. Leclercq et al. demonstrated that the hepatic expression of *Abcd2* and *Abcd3*, but not *Abcd1* and *Abcd4*, exhibits a high degree of sensitivity toward dietary PUFA intake [135].

#### 1.4.4. Peroxisomal FAO Related to the Synthesis of Cholesterol and Phospholipids

Acetyl-CoA formed during FAO in peroxisomes can be used for synthesizing cholesterol and phospholipids (mainly plasmalogen) [136]. For instance, the first two steps of plasmalogen biosynthesis occur in peroxisomes from the acetyl-CoA derived from peroxisomal FAO [137]. Recent studies indicate that peroxisomal βOX stimulates cholesterol biosynthesis in the liver of diabetic mice [138]. Moreover, it has been reported that the inhibition of peroxisomal βOX suppresses cholesterol biosynthesis and consequently lowers plasma cholesterol concentration. Based on these data, the authors suggest that the upregulation of peroxisomal cholesterol biosynthesis related to βOX may contribute to diabetes hypercholesterolemia [138].

#### 1.4.5. Peroxisomal FAO—Inhibition of Lipophagy

Lipophagy involves the encapsulation of lipid droplets into the autophagosome, which fuses with the lysosome, resulting in the hydrolysis of triacylglycerols catalyzed by lysosomal acid lipase A [110,139,140,141]. Peroxisomal FAO in the liver promotes hepatic steatosis by inhibiting lipophagy [141]. Supplied by FAO, acetyl-CoA is involved in the acetylation of Raptor, a component of mTORC1, a metabolic regulatory complex that inhibits autophagy [141].

#### 1.4.6. Peroxisomal FAO—Regulation of Mitochondrial β-Oxidation

Peroxisomal βOX increases the cellular NADH/NAD^+^ ratio, which inhibits the SIRT1/AMPK pathway. The inhibition of that pathway leads to increased ACC activity. It causes elevation of malonyl-CoA levels in the cytosol, inhibiting CPT1 and the transport of LCFAs into mitochondria, decreasing mitochondrial βOX [110,142].

#### 1.4.7. Peroxisomal FAO As a Process Associated with the Production of H_2_O_2_—An Important Signaling Molecule and Toxic Substance

As mentioned above, peroxisomal FADH_2_ formed during βOX is involved in H_2_O_2_ production. H_2_O_2_ is an important signaling molecule that regulates many cellular processes by modulating the activity of several proteins via cysteine oxidation [143]. Under physiological conditions, catalase converts most of the H_2_O_2_ formed during peroxisomal βOX to H_2_O and O_2_ [144]. However, when catalase activity is decreasing, for instance, during aging, part of H_2_O_2_ formed via peroxisomal βOX diffuses out the peroxisome (it is a relatively stable ROS) and may modulate the activity of redox-sensitive protein, which in turn triggers a complex network of signaling processes leading to regulation of (a) NF-ϰB activation, (b) E cadherin expression, (c) the secretion of matrix metalloproteinases, (d) mTORC activity, and (e) autophagy [144,145]. However, it is generally believed that reactive oxygen species (ROS) play a dual role. At physiological conditions, they are required for many signaling processes, affecting proliferation, differentiation, and aging, but there are also toxic byproducts of aerobic metabolism, including products of FFA oxidation [146]. H_2_O_2_ can be converted to highly reactive hydroxyl radicals, causing damage to proteins, lipids, and DNA, leading to many diseases, including atherosclerosis, cancer, diabetes, and rheumatoid arthritis [147]. Thus, it is tempting to speculate that microsomal βOX, via H_2_O_2_ production, may affect aging processes and aging-related diseases.

#### 1.4.8. Microsomal Fatty Acid ω-Oxidation

Under physiological conditions, FA ω-oxidation accounts for no more than 10% of total fatty oxidation in the liver [2]. In this process, the terminal methyl group (ω carbon) of FFAs is oxidized to the carboxyl group. The first step of ω-oxidation is catalyzed by the CYP family present in the microsome (including CYP4F2 and CYP4F3B), which requires NADPH and O_2_. Formed ω-hydroxy-FFAs are oxidized to ω-oxo-FFAs by cytosolic alcohol dehydrogenase. Finally, ω-oxo-FFAs are oxidized by cytosolic aldehyde dehydrogenase to carboxy-FFAs. Formed carboxy-FFAs (dicarboxylic-FAs) can be excreted into the urine or transported into mitochondria or peroxisomes, where they are metabolized via βOX. It should be noted that phytanic acid (described above) can also be oxidized via ω-oxidation [2]. Moreover, it has also been postulated that microsomal ω-hydroxylase is involved in (a) the synthesis of ω-hydroxylated arachidonic acid in the human liver and kidney, which regulates cardiovascular function (as vasoconstrictor), (b) ω-oxidation, and consequently the inactivation of leukotriene B4 (LTB4) in human leukocytes, and (c) the ω-oxidation of MCFAs and some xenobiotics [2].

## 2. The Function of FAO in Selected Organs

### 2.1. Liver

In the liver, FAO takes place in mitochondria and peroxisomes [148]. In a condition of low dietary carbohydrate supply or a prolonged fasting state, the activity of FAO increases significantly in the liver mitochondria, which is associated with a significant amount of energy production. In the liver, FAO is also the predominant source of acetyl-CoA, the substrate for ketone bodies (KBs) synthesis, and an important substrate for cholesterol synthesis, II phase detoxication, protein acetylation, and the synthesis of many other compounds, including N-acetylglutamate (NAG) synthesized by N-acetylglutamate synthase [149,150,151,152,153].

When intensive FAO occurs in the liver, acetyl-CoA and acetoacetyl-CoA (products of FAO) are used in the mitochondrial matrix to synthesize KBs. Acetoacetyl-CoA can also be formed by condensing two acetyl-CoA molecules in a reaction catalyzed by acetyl-CoA acetyltransferase 1. Subsequently, mitochondrial 3-hydroxy-3-methylglutaryl-CoA synthase 2 (HMGCS2) catalyzes the condensation of acetoacetyl-CoA and acetyl-CoA, generating 3-hydroxy-3-methylglutaryl-CoA (HMG-CoA), which is cleaved by HMG-CoA lyase (HMGCL), yielding acetoacetate (AcAc) and acetyl-CoA. AcAc can be reduced to D-β-hydroxybutyrate (BHB) by D-β-hydroxybutyrate dehydrogenase 1 (BDH1). In addition, AcAc can undergo spontaneous decarboxylation to acetone [82,154]. The plasma concentration of KBs under physiological conditions in humans is low (0.05–0.1 mmol/L) and significantly rises during prolonged starvation, ketogenic diet consumption, or insulin deficiency to 5–8 mmol/L and even 20 mmol/L in severe diabetic ketoacidosis [154]. Of the total pool of circulating KBs, BHB accounts for about 70% and is the most abundant [152,154]. The BHB to AcAc synthesized ratio is proportional to the mitochondrial NADH/NAD^+^ equilibrium [155]. The formed BHB and AcAc are alternative energy sources for extrahepatic tissues, particularly skeletal muscle, heart, kidneys, and the nervous system during diminished glucose availability [154,155]. The main regulatory steps of ketogenesis include (a) the availability of FFAs to hepatocytes, (b) the transport of acyl-CoA into mitochondria, and (c) HMGCS2 activity, a rate-limiting enzyme in ketogenesis. HMGCS2 is regulated at the level of gene transcription and by post-translational modifications [154]. The increased level of ketogenesis also occurs in subjects on a ketogenic diet and patients with severely uncontrolled diabetes [151,152]. KBs produced during ketogenesis are AcAc, BHB, and acetone [152].

#### 2.1.1. Mitochondrial FAO As a Regulator of Gluconeogenesis

In carbohydrate-deficient states, gluconeogenesis is the primary source of blood glucose. The stimulation of gluconeogenesis is attributed to mitochondrial FAO in connection with the production of acetyl-CoA and NADH. Acetyl-CoA is an allosteric activator of pyruvate carboxylase, a key gluconeogenic enzyme, whereas NADH is used to form 3-phosphate glyceraldehyde (precursor of glucose) from 1,3-bisphosphoglycerate [156]. Furthermore, the acetyl-CoA is an activator of pyruvate dehydrogenase kinase, which phosphorylates and consequently inhibits the pyruvate dehydrogenase complex (PDC), inhibiting the conversion of pyruvate into acetyl-CoA and further into the TCA [155,157]. Accumulating pyruvate can be converted by pyruvate carboxylase to oxaloacetate, a glucose precursor.

#### 2.1.2. Mitochondrial FAO As a Source of Acetyl-CoA for Protein Acetylation

Protein acetylation is a reversible post-translational modification of proteins, which involves the transfer of the acetyl group from acetyl-CoA to the ε-amino group of lysine [150]. Acetylation is catalyzed by lysine acetyltransferase using acetyl-CoA as one of the substrates (the second substrate is a non-acetylated protein). Acetylation can also occur non-enzymatically, and this process increases with increasing pH [150]. The acetyl-CoA necessary for acetylation is formed during mitochondrial βOX. It was shown that the hyperacetylation of liver protein depends on βOX since mice deficient in βOX cannot increase the acetylation of proteins [158]. Deacetylation is catalyzed by lysine deacetylase [150]. Protein acetylation and deacetylation are important regulatory mechanisms that modulate more than 2000 proteins (Figure 4) [159]. Interestingly, enzymes regulated by acetylation/deacetylation include FAO enzymes (LCAD and MCAD). LCAD is acetylated and consequently inactivated at lysine 42. Deacetylation and, consequently, the activation of LCAD is catalyzed by SIRT3—an NAD^+^-dependent protein deacetylase [160]. MCAD is acetylated and inactivated at lysine 318 and 322 [161]. It should be noted that liver mitochondrial enzymes regulated by acetylation and deacetylation are also (a) enzymes involved in ketogenesis and (b) enzymes involved in urea synthesis [162,163,164,165].

#### 2.1.3. The Potential Role of Mitochondrial FAO in the Regulation of Ureagenesis

Human adults produce approximately 1 mol (approx. 17 g) of toxic ammonia daily, most of which, via carbamoyl phosphate, is converted to nontoxic urea (at physiological concentrations) in the urea cycle [166]. It is well known that the synthesis of urea in humans and most animals requires, among others, four molecules of ATP per one formed molecule of urea as a source of energy and NAG as a positive allosteric activator of carbamoyl phosphate synthetase I (CPS1), a key regulatory enzyme in the urea cycle [167]. In theory, mitochondrial βOX may be involved in the production of both ATP and NAG. Indeed, some data indicate that an increase in liver FAO was associated with increased NAG level [168]. Therefore, stimulation of the liver FAO may likely increase acetyl-CoA level, a substrate for NAG synthesis and a key activator of CPS1. Thus, one can conclude that liver mitochondrial FAO plays an important role in the regulation of ureagenesis. In this manner, liver FAO appears necessary to prevent the accumulation of free ammonia, a neurotoxic compound, in blood and other tissues, including the brain. This conclusion is confirmed by published data, indicating that the defect of liver mitochondrial βOX is associated with hyperammonemia [169].

#### 2.1.4. The Potential Role of Mitochondrial FAO in Phase II Detoxication

The liver requires a lot of ATP to perform detoxication of xenobiotics and endogenously produced substances (for instance, the conversion of ammonia to urea described above). ATP is needed mainly to synthesize uridine diphosphate glucuronic acid, glutathione, 3′-phosphoadenosine-5′-phosphosulfate, and S-adenosylmethionine, compounds playing a key role in phase II of detoxication. Energy can be provided by βOX. Moreover, acetyl-CoA in phase II detoxication can be formed during liver mitochondrial βOX. N-acetyl transferases (NATs), also known as arylamine N-acetyl transferases, play an important role in the phase II detoxication of xenobiotics, including drugs and detoxication [153]. In humans, the acetylation of xenobiotics is catalyzed by NAT1 and NAT2. These enzymes are responsible for transferring the acetyl group from acetyl-CoA to convert aromatic amines to aromatic amides or hydrazines to hydrazides [153]. It should be noted that in humans, acetylation is an important route of biotransformation for many arylamine and hydrazine drugs, as well as for the biotransformation of carcinogens present in the diet, cigarette smoke, and environment.

#### 2.1.5. Hepatic Manifestations of FAO Disorders (FAOD) Caused by Genetic Defects

One of the frequent manifestations in patients with FAOD is liver dysfunction. It is mainly associated with deficiencies of VLCAD, LCHAD, MCAD, CPT1, CPT2, and CAC [82,170]. Symptoms are triggered by extended fasting, exercise, fever, sepsis, and other metabolic stress causes. Liver dysfunction resulting from abnormal FAO usually appears early in life. It may include hypoketotic hypoglycemia or liver dysfunction resulting from hepatocyte damage, including Reye syndrome. Hepatic symptoms may also occur later in life [170]. Hypoglycemia in patients with FAOD occurs when glycogen stores are depleted, possibly due to increased peripheral glucose uptake. It may result from impaired energy production from FFAs and the reduced synthesis of KBs by the liver [171]. It may also be a consequence of reduced hepatic gluconeogenesis [172]. FAOD can lead to the sudden death of newborns, mainly due to the limited glycogen reserves and high metabolic rate [171]. Most of the liver damage observed in FAOD is due to the toxic effects of accumulating FFAs and their carnitine derivatives. These toxic effects are related to (a) the inhibition of the mitochondrial respiratory chain and energy deficiency, (b) increasing reactive oxygen species (ROS) formation, and an imbalance in calcium homeostasis, leading to mitochondrial damage and further apoptosis or necrosis of cells [171,173]. Symptoms of Reye-like classified hepatic-presenting FAOD include hepatic encephalopathy, hepatomegaly, hyperammonemia, and microvesicular steatosis of the liver [170].

### 2.2. Heart and Skeletal Muscles

The heart requires enormous quantities of ATP to maintain its contraction capacity and ion homeostasis. ATP and phosphocreatine stored in cardiomyocytes ensure the heart works properly for only a few seconds. Therefore, ATP must be constantly synthesized (from ADP and Pi), mainly through oxidative phosphorylation, providing approx. 95% of ATP, with anaerobic glycolysis providing the rest. A healthy subject’s heart is metabolically flexible and readily shifts between energetic substrates [174]. In the resting state, βOX contributes to the synthesis of approx. 50–70% of ATP. The remaining is mainly provided by glucose oxidation. KBs (mainly BHB) are the third supplier of ATP (10–15%), whereas amino acids contribute 1–2% for energetic purposes [175]. During exercise or myocardial stress, lactate may also be an important fuel for the myocardium [176].

Significant changes in heart mitochondrial energy metabolism are related to pathological conditions. In diabetes, the ratio of FAO to glucose oxidation is increased due to elevated FAO and lowered glucose oxidation [177]. The inhibition of glucose utilization by FFAs occurs at multiple levels, including glucose uptake by cells, the rate of glycolysis, and mitochondrial oxidation. Recent research suggests that cardiac metabolic overload with oleate or palmitate leads to increased global protein acetylation, which inhibits glucose transporter type 4 (GLUT4) translocation to the plasma membrane and consequently inhibits glucose uptake [178]. Lipid abnormalities leading to atherosclerotic plaque formation in the vascular wall also induce a remodeling of the energy metabolism in cardiac myocytes toward accelerated FFA and branched-chain amino acid oxidation. Redirection toward FAO increases the oxygen cost of ATP formation and may become maladaptive and reduce myocyte survival rates under acute oxygen deprivation [179]. The administration of trimetazidine (a competitive inhibitor of LCKAT), etomoxir, or perhexiline (inhibitors of CPT1) resulted in a cardioprotective effect in humans with heart failure (HF), probably through the inhibition of FAO and an increase in glucose oxidation [174,180]. In general, these data suggest that reduced FAO might improve cardiac function under pathological conditions.

However, the downregulation of LCAD or MCAD in patients with HF and animals during HF progression was detected. Moreover, impaired FAO contributes to the progression of HF by altering cardiac energy metabolism after myocardial infarction [181]. Thus, the problem of whether a reduction in FAO improves or worsens cardiac function is still an open question.

Inconsistent results have also been reported regarding the level of malonyl-CoA (as a natural inhibitor of FFA oxidation) and its role in cardiac function. The inhibition of MCD, increasing the malonyl-CoA level (inhibitor of CPT1, and consequently FAO), improved cardiac function by increasing cardiac output. The promising results of MCD inhibition were associated with the reduced production of protons due to enhanced coupling between glycolysis and glucose oxidation [182]. However, ACC inhibition, resulting in a decrease in the malonyl-CoA level, which stimulates the oxidation of FFAs, was also associated with a cardioprotective impact in the failing mouse heart [183].

Animal studies showed that increased FAO (caused by *ACC2* deletion) did not induce cardiac dysfunction [184]. In addition, it was demonstrated that increased FAO in the heart protects against cardiomyopathy in chronically obese mice [183]. However, a strong correlation between decreased cardiac efficiency and an over-dependence on FAO has been reported in ob/ob mice and obese humans [185,186].

Increased cardiac FAO has been considered to cause elevated ROS production in mitochondria and subsequent oxidative damage of mitochondria, contributing to cardiac dysfunction in obese rodent models [187,188]. The molecular mechanisms responsible for FAO-induced lipotoxic cardiomyopathy are also unclear [184]. Several pathogenetic pathways have been proposed, such as mitochondrial dysfunction and oxidative stress, ER stress, and apoptosis induced by toxic lipids.

Levels of acyl-CoA are reduced in failing human hearts and hypertrophic mouse hearts. The heart-specific *ACSL1* overexpression in mice causes an increase in acyl-CoA levels and a stable turnover of TAG with the preservation of all cardiac functions after pressure overload surgery. Therefore, it was suggested that therapies aimed at enhancing or mimicking the effects of ACSL1 could positively impact the treatment of chronic HF [189].

Cardiac dysfunction due to inborn errors in LCHAD, MTP, neonatal CPT2, VLCAD, and MCAD is the most common [190,191]. This FAOD may manifest in the neonatal period with severe symptoms, including cardiomyopathy, hepatic dysfunction, and hypoketotic hypoglycemia.

Patients with FAOD may develop hypertrophic cardiomyopathy due to an inadequate energy supply to the heart and the subsequent inefficient contraction [170]. Arrhythmias in FAOD patients are often multifactorial but mainly occur as LC-FAO defects. Conduction disturbances and atrial tachycardia were detected in patients with CPT2, CAC, and LCHAD/MTP deficiency [192]. Ventricular tachycardias were observed in patients with FAO deficiency [193]. It is critical to quickly and correctly identify significant signs and symptoms in patients with FAOD to manage metabolic decompensation and reduce possible comorbidities. Cardiac arrhythmias and hypoglycemia are often observed in the early postnatal period and may lead to sudden infant death syndrome. Therefore, inborn errors of FAO should be considered in all instances of sudden unexplained death [170]. In infancy and early childhood, FAOD may manifest as cardiac, skeletal muscle, and liver dysfunction and may also cause fasting or exercise-induced hypoketotic hypoglycemia, Reye-like syndrome, cardiomyopathy, and recurrent rhabdomyolysis [190]. Muscular symptoms, especially rhabdomyolysis and cardiomyopathy, are most common in adolescents or adults [194].

Heart failure associated with FAO deficiency is difficult to treat. Moreover, available treatments need to address the fundamental pathologies of LC-FAODs. Using medium even-chain triacylglycerols (MCT oil), which provided the MCFA source (mainly octanoate), did not eliminate symptoms of LC-FAO defects due to a deficit of TCA intermediates [190]. Triheptanoin (UX007, Ultragenyx Pharmaceuticals) is a triacylglycerol composed of seven carbon (C7:0). It was reported that the oral administration of triheptanoin resulted in a significant and rapid beneficial effect on cardiac function in children with various genetic FAO disorders (VLCAD, MTP, LCHAD, or CAC deficiency) [195]. Vockley et al. demonstrated after a long-term study that triheptanoin treatment was associated with significant improvements in glucose homeostasis and cardiomyopathy. Moreover, episodes with rhabdomyolysis were also reduced but with less effect than the other symptoms, which may suggest different pathophysiologic mechanisms that require additional therapy [196,197].

FAODs with skeletal myopathy occur most frequently in LCHAD, MTP, VLCAD, and CPT2 defects. Lack of energy production during the FAO process in skeletal muscles results in fatigue, which manifests as myalgia, muscle weakness, myoglobinuria, physical intolerance, and episodes of rhabdomyolysis. Myopathy usually begins due to excessive endurance exercise, anesthesia, or a viral illness in adolescents or adults but can also appear earlier. A significant deficiency of ATP in muscle cells leads to rhabdomyolysis, which, consequently, causes the release of myoglobin into the extracellular fluid and circulation [190]. It was demonstrated that bezafibrate, a PPARα agonist, might reduce rhabdomyolysis episodes in patients with CPT2 deficiency [198]. However, a different study demonstrated no beneficial effect of bezafibrate on FAO or physical ability [199]. Due to the absence of highly effective therapies to prevent rhabdomyolysis associated with FAO, patients with these disturbances should reduce prolonged and intense physical activity.

Increased skeletal muscle FAO has been proposed as a potential mechanism leading to impaired muscle insulin sensitivity [200]. Gavin and colleagues revealed that patients with poorly controlled type 2 diabetes (T2D) have elevated incomplete skeletal muscle FAO compared with well-controlled T2D patients [201]. Moreover, incomplete FAO was inversely related to muscle insulin sensitivity and glycemic control. The experiment also indicated that elevated HbA1c is associated with the upregulation of FAO gene expression in the skeletal muscle of T2D patients. Lipid overloading promotes incomplete FAO, increasing acylcarnitine levels in T2D patients’ plasma, possibly resulting in insulin resistance.

FAO is also dysregulated in the skeletal muscles of obese individuals. Several studies comparing metabolism in the muscles of obese and lean individuals demonstrated that in obesity, the skeletal muscle metabolic capacity is primarily involved in FA esterification and storage rather than oxidation [202,203]. In the skeletal muscle of obese women, maximal CPT1 activity was decreased by 27–35% compared to lean women. Moreover, the ratio of muscle CPT1 activity to FABPm protein in obese individuals was half the level detected in lean individuals [204]. This may suggest that in obesity, FAs can be taken up from plasma but cannot be further used as an energy source due to the muscle-reduced capacity for FA oxidation. Aerobic exercises seem appropriate to improve FAO and lipid metabolism in healthy and insulin-resistant obese individuals [203].

### 2.3. Kidney

Removing waste from the blood, reabsorbing glucose and other nutrients, regulating the balance of electrolytes and fluid, maintaining acid-base homeostasis, and regulating blood pressure by the kidney requires the continuous synthesis of ATP. FFAs serve as key substrates for energy production in the kidney [205]. Low βOX may contribute to the development and progression of kidney diseases due to low ATP levels and the excessive accumulation of triacylglycerols, leading to cellular lipotoxicity and the development of tubulointerstitial fibrosis [206,207,208]. The proximal tubule cells prefer FAO over glycolysis as a process of synthesizing ATP and display low metabolic flexibility between FAO and glycolysis, which make these cells more sensitive to acute and chronic hypoxia [209,210]. In contrast, the distal tubule cells are less susceptible to ischemic injury and nephrotoxins because they may switch from FAO to glycolysis during hypoxic/ischemic conditions [210].

The system of delivering FFAs to kidney cells is generally similar to other organs (presented above). Briefly, FFAs can be taken up by the proximal tubular cells by special FFA transporters (CD36, FABPs, FATPs) or reabsorbed from the glomerular filtrate by the endocytosis of receptor-mediated FA-bound albumin [210,211].

The downregulation or deficiency of CPT is crucial to impaired FAO in experimental models of acute kidney injury or diabetic nephropathy [210,212,213]. It has been shown that impaired lipid metabolism may be linked directly to kidney fibrosis [212,214]. Usually, kidney fibrosis is associated with the transforming growth factor (TGF-β) and is the final pathological process of any ongoing chronic kidney disease (CKD) or maladaptive repair. The changes in *CPT1* expression significantly ameliorated FAO metabolism in the kidney [212]. Patients with CKD present decreased activity of CPT1 and an increased accumulation of short- and middle-chain acyl-carnitines due to impaired FAO. Therefore, strategies that can improve the mitochondrial structure and function, overcome the negative effect of TGF-β on the oxygen consumption rate, and promote tubular epithelial cell differentiation are postulated as potent therapeutics for kidney fibrosis in CKD [212,215]. In general, TGF-β takes part in many physiological and pathological processes, including (a) angiogenesis, (b) apoptosis, (c) the division of mesenchymal cells, (d) the regulation of the synthesis and the degradation of extracellular matrix protein. At the molecular level, TGF-β1 inhibits the expression of *CPT1* and decreases FFA catabolism. Moreover, TGF-β1 also represses the synthesis of mRNA encoding the upstream regulators of CPT1, namely PPARα and PPARγ coactivator-1α (PGC1α) [216,217]. Genome-wide transcriptome studies revealed that enzymes and regulators of FAO are reduced in the kidneys of patients with CKD and experimental models of kidney fibrosis [217]. Mice with kidney injury treated with etoxomir (a specific inhibitor of CPT1) display a higher expression of fibrosis markers [218]. In addition, treating mice with C75, a synthetic compound that increases CPT1 activity, decreases the apoptosis rate in the kidney [217]. The above-presented data suggest that CPT plays a key role in kidney physiology and pathology.

It has been postulated that restoring FAO by regulating the level or activity of PPARα and TGF-β may improve the treatment of kidney disorders [219]. PPARs and PGC1α are the critical transcription factors/coactivators that regulate the expression of proteins involved in the uptake and oxidation of FFAs [220]. The administration of fenofibrate, the agonist of PPARα, strongly induces the expression of genes encoding FAO enzymes (*Cpt1*, *2* and *Acox1*, *2*). Mice with kidney insufficiency injected with fenofibrate demonstrated a decreased expression of caspase 3, a reduced apoptosis rate, reduced fibrosis, reduced kidney injury, and improved renal function. This suggests that fenofibrate treatment restores FAO-related enzyme expression and may prevent lipid metabolism abnormalities in kidney diseases [217,220]. The protective effect of Wy-14643 (the PPARα ligand) was also demonstrated in cisplatin-induced renal failure. Cisplatin causes a significant reduction in proximal tubule FAO. PPARα ligands prevent acute tubular necrosis by ameliorating the cisplatin-induced inhibition of two distinct metabolic processes, MCAD-mediated FAO and PDC activity [219,221]. Also, the ketogenic diet enhanced FAO in mice with kidney fibrosis, reducing fibrosis in this organ [222]. Overall, βOX provides enough energy to support various kidney functions and ensures the kidney’s structural integrity [223].

### 2.4. Lungs

Recent studies indicate that βOX can also play an important role in pulmonary fibrosis, especially idiopathic pulmonary fibrosis (IPF), a fatal fibrotic disorder of unknown etiology [224]. Increased activity of FAO was observed in IPF lungs, which suggests that βOX can be involved in fibrinogenesis, mainly via macrophage activation [225]. Furthermore, βOX provides ATP, which is believed to promote macrophage M2 polarization, which plays a key role in fibrogenesis [226]. It has also been shown that macrophage CD36, involved in FFA transport, plays an important role in fibrogenesis since the loss of CD36 inhibits lung fibrosis [227]. Overall, the data presented above indicate that FAO can play an important role in developing IPF.

### 2.5. Enterocytes and Colonocytes

Glutamine and glutamate are the main energetic substrates for enterocytes. However, enterocytes can also oxidize FFAs entering the cells from the plasma and intestinal lumen. FFAs derived from the intestinal lumen (directly derived from dietary lipids, mainly TAG) provide more energy to enterocytes (approx. 60%) than FFAs derived from the plasma (approx. 40%) [228]. A high-fat diet significantly induces FAO in enterocytes. However, when animals are fed a high carbohydrate diet, FFAs are not an important energy source for enterocytes. It has been proposed that in addition to energy production, FAO in the small intestine (enterocytes) could be a sensor that affects eating behavior [228]. However, further studies are required to confirm this suggestion.

Colonocytes mainly oxidize SCFAs, including acetate, propionate, and butyrate, which are produced by gut microbiota. Butyrate is the main energy source of colonocytes and uses more than 70% oxygen for butyrate oxidation [229]. Any impairment of SCFA oxidation leads to a disturbance in colonocyte function. For instance, it has been shown that reducing SCFA oxidation by ibuprofen (a nonselective and nonsteroidal anti-inflammatory drug) may cause an ulcerative [230].

### 2.6. βOX in Other Organs/Tissues/Cells

#### 2.6.1. Adipocytes

It was suggested that increased FAO in adipocytes might be a promising therapeutic strategy for chronic inflammatory diseases, including obesity and T2D [231]. An experiment with chickens revealed that fasting rapidly increases FAO in white adipose tissue (WAT) by upregulating the expression of genes involved in this process. Enhanced oxidation precedes the high level of FFAs in serum, indicating that FAO is induced at the early stages of lipolysis. Therefore, it may act as an adaptive response to elevated intracellular FFA levels in adipocytes [232]. Gonzalez-Hurtado et al. demonstrated that FAO is critical not only for adipose bioenergetics but also for the browning of WAT and BAT survival under acute thermogenic activation and during periods of BAT quiescence [233].

#### 2.6.2. Brain

It is generally believed that glucose and KBs during starvation, but not FFAs, are energy substrates for the brain. It has been suggested that a lack of active βOX in neurons may protect these cells against excessive ROS production and hypoxia [234]. As was already discussed, both processes’ intensity (excessive ROS production and hypoxia) increase in the cells oxidizing FFAs. However, some recent studies indicate that βOX can provide up to 20% of the energy used by the entire rat brain [235]. Moreover, it has been shown that FFAs can be transported through the blood–brain barrier and oxidized by astrocytes [236,237,238,239]. Acetyl-CoA, formed as the end product of the βOX in astrocytes, can be used as a substrate for KB production. Formed KBs can be transported to neurons, where they serve as an energy substrate [240]. Additionally, FFAs that are peroxidized in hyperactive neurons can be transported to astrocytes and stored in lipid droplets or oxidized in βOX [241]. Our recent review extensively discussed the function of FAO in the brain [242].

#### 2.6.3. Endothelium

Endothelial cells (ECs) produce more than 85% of the energy needed in anaerobic glycolysis [243]. However, it was demonstrated that in proliferating ECs, acetyl-CoA produced during βOX contributes a significant portion of the carbons required for the TCA intermediates—precursors of substrates necessary for de novo dNTP synthesis [243,244]. Furthermore, Kalucka et al. demonstrated that quiescent ECs upregulate FAO enzymes to maintain the TCA for redox homeostasis through NADPH by isocitrate dehydrogenase 2 (IDH2) and ME3 [245]. Summing up, one can say that βOX takes place in ECs and plays an important role in some processes, including de novo dNTP synthesis and maintaining redox homeostasis.

#### 2.6.4. Placenta

A very early work from our department demonstrated palmitoyl–carnitine oxidation in mitochondria isolated from the human term placenta [246]. Later, it was demonstrated that FAO enzyme activity in the human placenta was higher early in gestation and lower in term [247,248]. Moreover, it has been shown that a deficiency in FAO may result in placental dysfunction, leading to gestational complications [249]. An increased expression of genes associated with βOX has been observed in the human placenta in pre-eclampsia [250]. Recent studies also indicate the important role of βOX in the placenta for normal fetal development, although the expression of genes related to βOX in the term human placenta is about 20 times lower than in the liver [251,252]. Recently published data indicate that human placental FAO can be inhibited by high glucose concentration in pregnant women with diabetes. Based on these data, it has been suggested that inhibiting FAO can lead to an increase in lipid transfer to the fetus and, consequently, excessive fetal growth [253]. The results presented above suggest that FAO plays an important role in developing the human placenta and the normal course of pregnancy.

#### 2.6.5. Peripheral White Blood Cells

Glycolysis and glutaminolysis provide enough ATP for the normal function of peripheral white blood cells [254]. However, it has been shown that FFAs are also oxidized by human white blood cells. Moreover, it has been demonstrated that βOX is not significantly affected by sex or acute exercise, but genetic factors play a significant role in determining the level of FAO [255]. Interestingly, in healthy subjects’ peripheral blood cells, specific carnitine esters (different from other tissue) are accumulated [256]. Accordingly, different amounts and patterns of acylcarnitine esters were found in patients with defects of βOX [256,257]. It may have practical significance since analyzing βOX intermediates in peripheral blood cells may allow the identification of FAO defects.

#### 2.6.6. Steroidogenic Cells

It has been shown that FAO is also active in steroidogenic tissues. Moreover, it has been demonstrated that FAO activity in steroidogenic cells is regulated by translocator protein (TSPO), also known as the peripheral benzodiazepine receptor [258]. This protein is located in the outer mitochondrial membrane, and its depletion leads to increased (a) FFA uptake by mitochondria, (b) FAO, and (c) ROS production. TSPO depletion in cells induces a shift in substrate oxidation from glucose to FFAs for energy production. The authors suggest that TSPO can play an important role in modulating FAO not only in steroidogenic tissue but also in cells active in lipid storage and metabolism [258].

#### 2.6.7. Osteoclast

Bone formation by osteoblasts and bone resorption by osteoclasts play a crucial role in skeletal remodeling. These processes require a large amount of ATP produced by glucose, FA, and amino acid oxidation [259,260]. Several years ago, it was shown that active osteoclasts exhibit HAD activity [261], suggesting that βOX takes place in these cells (active osteoclast). Some data indicate that βOX is involved in osteoclastogenesis [262]. It has also been shown that the cell membrane of osteoclast possesses transporters involved in LCFA uptake [263,264]. Moreover, it has been reported that the high energy state of an active osteoclast (osteoclast in the active bone resorption state) could be supported by lipid catabolism [265].

Recent studies showed (a) a significant increase in LCFA oxidation during osteoclast differentiation. This was associated with increased mRNA and protein levels of enzymes involved in βOX [266]. Thus, mitochondrial FAO is important for normal osteoclast formation and function. Based on these data, one can conclude that FFAs are key energy sources necessary for bone remodeling, and their inhibition may lead to a disturbance in osteoclast formation and function [266]. For instance, some authors suggest the role of osteoclast energy metabolism in the development of osteoporosis [260]. Very recently, the upregulation of CPT1A and increased FAO in osteoclast precursors of patients with rheumatoid arthritis has been shown [267]. Moreover, enhanced FAO influences osteoclastogenesis and promotes cell–cell fusion during osteoclast maturation. In contrast, the knockdown of the *CPT1A* gene or the inhibition of CPT1A activity by etomoxir (pharmacological inhibitor of CPT1A) blocked osteoclastogenesis. Based on these data, the authors conclude that increasing FAO in osteoclast precursors participates in joint destruction in patients with rheumatoid arthritis [267]. The results presented above indicate that FAO plays an important role in providing energy for osteoclastogenesis and, consequently, skeletal remodeling. Disturbance in FAO in active osteoclasts might lead to osteoporosis, whereas osteoclast precursors lead to joint destruction in rheumatoid patients.

#### 2.6.8. Pancreatic β-Cell

FAO in the pancreatic β-cell is involved in the regulation of insulin secretion [268]. Many years ago, it was shown that FFA catabolism via mitochondrial βOX is an important energy source for pancreatic β-cells [269]. It is also well known that energetic substrates, mainly glucose, regulate insulin secretion by pancreatic β-cells. However, FFA and amino acids also stimulate glucose-induced insulin secretion [270]. Glucose metabolism plays a crucial role in the stimulation of insulin secretion by pancreatic β-cells. It is generally believed that glucose metabolism in pancreatic β-cells (via a sequence of the following events: an increase in the ATP/ADP ratio → closure of the ATP-sensitive K channels → the cell membrane depolarization and opening of voltage-sensitive Ca^2+^ channels) raises intracellular Ca^2+^ concentration and triggers exocytosis of insulin-containing granules [271]. FFAs have also been shown to stimulate glucose-induced insulin secretion by pancreatic β-cells over short-time exposure [271]. However, the mechanism by which FFA may stimulate insulin secretion by pancreatic β-cells is still unknown. By combining several data, Prentki et al. created a comprehensive model called the “trident model of pancreatic β-cells lipid signaling” to explain the role of FFAs in stimulating insulin secretion by pancreatic β-cells [272]. In a nutshell, the model takes into account three interdependent processes. Two of them are strictly related to the intracellular metabolism of FFAs and the third is related to membrane FFAR (the free fatty acid receptor present in pancreatic β-cells) activation. The first intracellular process proposed in this model is associated with elevated levels of LC-CoA in pancreatic β-cells. It occurs via a sequence of the following events: glucose metabolism (glucose → → pyruvate → acetyl-CoA → malonyl-CoA), which leads to an increase in malonyl-CoA, which inhibits CPT1 and consequently slows down FAO. As a consequence of FAO inhibition, an intracellular increase in LC-CoA takes place. LC-CoA regulates many pancreatic β-cell functions, including (a) the activation of some types of protein kinase C (PKC), which plays a crucial role in glucose-stimulated insulin secretion by pancreatic β-cells, (b) the modulation of ion channels (also involved in insulin secretion), the modulation of protein acylation channels (also involved in insulin secretion), and (d) the regulation of some gene transcriptions [271]. The second intracellular process of the trident model is associated with glucose metabolism, which (a) promotes FFA esterification by providing glycerol 3-phosphate and malonyl-CoA (as a physiological regulator of CPT1; see discussion above) and lipolysis (providing FFA), leading to an increase in intracellular DAG and phospholipids levels in pancreatic β-cells. Increased intracellular DAG and Ca^2+^ lead to insulin secretion by pancreatic β-cells mediated by PKC [271]. The third mechanism of the postulated trident model is associated with the binding and activation of FFAR1 (GPR40) by FFAs, which causes an increase in intracellular Ca^2+^, leading to insulin secretion by pancreatic β-cells. As mentioned, all these complex processes (two intracellular and one extracellular) stimulate insulin secretion by pancreatic β-cells.

The effect of FFAs on insulin secretion by pancreatic β-cells depends on exposure time, concentration, and the type of FFA [271,273]. Acute exposure caused an increase, whereas chronic exposure caused the suppression of insulin secretion by pancreatic β-cells [271]. Interestingly, mainly saturated FFAs (palmitate and stearate) synergize with elevated concentrations of glucose to cause pancreatic β-cell death (lipotoxicity), whereas oleate is practically nontoxic [273]. One possible explanation of the unfavorable effect of saturated FFAs on insulin secretion by pancreatic β-cells could be the negative regulation of *Idx-1* by saturated FFAs and the suppression of genes transactivated by IDX-1, including GLUT2, glucokinase, and insulin [274]. The inhibitory effect of FFAs (palmitate) strictly depends on βOX since it was prevented by inhibiting CPT1 [274].

Overall, the results discussed above indicate that mitochondrial βOX occurs in pancreatic β-cells and plays an important role in regulating insulin secretion.

In the pancreatic β-cells, similar to other organs, FAO occurs in mitochondria and peroxisomes [275,276]. However, it is not known to what extent peroxisome FAO contributes to FFA oxidation in pancreatic β-cells. Nevertheless, one has to remember that catalase, which is responsible for potentially toxic H_2_O_2_ (formed during peroxisome βOX) degradation, is practically not detectable in pancreatic β-cells, which might contribute to the development of T2D due to increased plasma FFA concentrations [276]. Moreover, it has been shown that the overexpression of catalase in the peroxisomes (but not in mitochondria) of insulin-producing cells (RINm5F cells with low catalase activity and good model cells for the study of H_2_O_2_-mediated lipotoxicity) (a) decreased the H_2_O_2_ level and (b) protected the cells against FFA-induced toxicity. Based on these data, it was postulated that peroxisomal βOX is involved in lipotoxicity via the synthesis of H_2_O_2_ [276].

## 3. FAO in Cancer

One of the distinctive features of cancer cells is a significant increase in ATP production. In cancer cells, ATP is needed to synthesize many micro- and macromolecules (often called biomass) that are essential for cell division and proliferation [277]. In most cancer cells, an increase in ATP synthesis is associated with an increase in glycolysis and glutaminolysis [278]. However, carcinogenesis is also related to significant lipid metabolism disturbances [279,280,281]. An upregulation of FAO enzymes has been reported in many malignancies [282,283,284,285,286,287,288]. The data presented in Table 4 indicate that gene-encoding FAO enzymes or proteins associated with FAO (e.g., FABPs) are upregulated in many, but not all, human cancers.

On this basis, it is conceivable that under conditions in which cancer cells require an additional amount of ATP, FAO can play an important role in ATP synthesis. Indeed, it has been shown that activated FAO increases ATP levels and promote cell survival in breast cancer cells and other tumor cells [337,338,339,340]. Moreover, it has been reported that CPT1C promotes cell survival and tumor growth under conditions of metabolic stress [313]. On the other hand, the inhibition of CPT1 resulted in a reduced proliferation of many cancer cells [341]. All results mentioned above indicate the important role of FAO in various cancer cells’ survival and growth. A potential role of FAO in cancer cell survival and growth is presented in Figure 5. As shown in Figure 5, FAO can provide not only ATP but also NADPH, an important compound for cancer cells’ growth and survival.

In cancer cells (similar to noncancer), NADPH is required for the generation of new building blocks, mainly FFAs (necessary for membrane phospholipids synthesis) and cholesterol (an important element of cells membranes), to sustain cell growth and proliferation [278,279,342]. Moreover, NADPH is also used to maintain cellular redox potential, mainly to keep a physiological level of reduced glutathione (GSH). GSH is a scavenger of toxic oxidative metabolites in the cancer cells and is involved in the conversion of excess harmful H_2_O_2_ to H_2_O. A disturbance in NADPH production in the cells increases sensitivity to ROS and, consequently, cell death [343]. Overcoming metabolic stress is an important process for tumor cell growth. Indeed, it has been shown that FAO may provide NADPH for defense against oxidative stress and glioblastoma cell death [344]. Similar results have been obtained using lymphoma cells (a subset of diffuse large B cells) and other carcinoma cells [313,339,340,342]. Therefore, increased NADPH production associated with FAO enhances redox buffering capacity and consequently protects cancer cells from ROS-induced damage. Overall, the data discussed above and presented in Figure 5 indicate the relevance of FAO for some cancer cell functions associated with ATP and NADPH production.

Several other data also suggest the contribution of FAO to cancer cell function. For instance, the uptake of FFAs from surrounding adipocytes promoted FAO in breast and colorectal cancers [345,346,347]. Moreover, some studies suggest that increased FAO may promote cancer metastasis by increasing ATP levels, allowing cancer cells to avoid apoptosis and facilitating epithelial-to-mesenchymal transition [341]. Recent studies reported that osteopontin, protein secreted by many cells, including adipocytes, upregulates the expression of *CPT1A* in prostate cancer tumor cells. The knockdown of *CTP1A* diminishes prostate cancer cells’ proliferation and invasiveness capacity. Furthermore, patients with the highest osteopontin gene (*SPP1*) expression had the worst prognostic outcome [348]. Some FAO genes were also altered in glioblastoma multiforme (GBM), the most aggressive brain cancer in adults [349]. The expression of *CPT1A*, *CPT1B*, and *ACAD9* was elevated in recurrent gliomas compared to primary tumors, whereas there was no difference in the expression of *VLCAD* and *SCAD* between primary and recurrent GBM. Moreover, the overexpression of *CPT1B*, *LCAD*, and *MCAD* was associated with lower overall survival of patients with GBM [289]. Various *ACSL* isoforms are overexpressed in colorectal, breast, prostate, and other cancers [290,350].

FAO may also increase the drug resistance of cancer cells, which was proven for dexamethasone, L-asparaginase, and tamoxifen [351,352,353]. Moreover, FAO may be essential in the chemoresistance and radioresistance of GBM and triple-negative breast cancer [208,289].

Moreover, the *LCAD* expression level was proposed as a hepatocellular carcinoma (HCC) patient mortality predictor [334,354]. The overexpression of *ACSL* in tumors of colorectal cancer patients is associated with a poorer prognosis [355]. Overall, one can conclude that FAO could be involved in invasiveness capacity, chemoresistance, and radioresistance, the promotion of cancer metastasis in some cancers, and be a mortality predictor.

The above-presented data suggest that FAO could be a potential therapeutic target, and its inhibition may reduce cancer cell proliferation, metastasis, and drug resistance. Table 5 presents examples of cancer cell FAO as potential therapeutic targets.

Using different cell lines, attempts have been made to inhibit the transformation at the stage catalyzed by ACSL. The inhibition of ACSL in cancer cells is associated with cell growth inhibition (Table 5). As CPT1 is the rate-limiting enzyme of FAO, most studies were looking for potential anticancer drugs focused on this enzyme. In mice with colon adenocarcinoma, the administration of etomoxir, an irreversible pharmacological CPT1 inhibitor, significantly delayed tumor growth and induced apoptosis [365]. It was also shown that inhibiting FAO by etomoxir enhanced the anticancer effect of cisplatin in HCT116 colon cancer cells [372]. Combining etomoxir with radiotherapy improved its effectiveness in an in vitro lung epithelial and prostate cancer cell model [366]. Moreover, some data indicate that CPT1 inhibition may prevent metastasis [288]. However, high concentrations of etomoxir can also inhibit complex I of the mitochondrial respiratory chain and reduce cell proliferation independently of FFA oxidation [373]. It should be noted that a more selective CPT1 inhibitor, teglicar, was developed, which is a reversible CPT1 inhibitor with less toxicity than etomoxir that prevented MYC-driven lymphomagenesis [374]. Perhexiline is an inhibitor of the CPT1 and CPT2 isoforms, and its use sensitizes cancer cells to the anticancer effect of oxaliplatin and increases their apoptosis [370]. Like other CPT inhibitors, perhexiline inhibits FFA oxidation, and enhanced ROS accumulation allows classical chemotherapeutic drugs to kill more CRC cells [370]. The results presented above and summarized in Table 5 indicate that FAO inhibitors have a potential role in cancer therapy. Importantly, some compounds presented in Table 4 (for instance, perhexiline, an inhibitor of CPT1, is approved for human use for the treatment of some diseases [278]). Therefore, these findings may represent an important step toward improving some cancer treatments in the near future.

It has been shown that a ketogenic diet or fasting limits tumor progression by different mechanisms, such as (a) lowering blood glucose and insulin concentrations, altering lipid metabolism, and (c) increasing BHB concentrations [375,376,377]. The recently published result indicates that the inhibition of succinyl-CoA:3-oxoacid-CoA transferase (SCOT), which plays a crucial role in KB oxidation, also reduces tumor volume and inflammation in the Lewis cancer model [378]. The reaction catalyzed by this enzyme is presented below:acetoacetate + succinyl-CoA → acetoacetyl-CoA + succinate

It suggests that KB oxidation can increase ATP production for the growth of cancer cells. Thus, one can suppose that FAO via an increase in KB synthesis may support cancer growth. The inhibition of SCOT may cause (a) a decrease in ATP synthesis and (b) an increase in BHB (precursor of acetoacetate) concentrations. Both a decrease in ATP synthesis and an increase in BHB may limit tumor progression by a different mechanism (BHB via Hcar2-Hops signaling) [377].

Together, the results presented above suggest that FAO may promote tumor growth, whereas the inhibition of FAO can lead to a reduction in tumor growth.

However, it should be emphasized that the involvement of FAO in cancer cells’ growth and function is still a debated issue because some data suggest that FAO is not necessarily relevant for ATP synthesis in certain cancer cells. The data presented in Table 4 indicate that gene-encoding FAO enzymes are downregulated in some human cancers. For instance, in HCC with a high *ACSL* expression, most genes encoding enzymes involved in FAO were significantly downregulated [293]. Similarly, in vitro and in vivo studies suggested that the downregulation of *MCAD* and *LCAD* enhances tumor proliferation and aggressiveness. Also, *ACSL1* is reported to be downregulated in non-small-cell lung cancer [290].

## 4. The Pathogenic Genetic Make-Up of FAO Genes

The diseases caused by mutations in gene-encoding FAO enzymes are rare or even very rare (for details, see Appendix A). Pathogenic changes may include sequence or copy-number variants. Some variations in FAO genes are part of more significant genetic disturbances. Associations between specific single-nucleotide polymorphisms (SNPs) in FAO genes and various biological traits or pathological conditions like T2D, cardiovascular disease, and CKD (for details, see Appendix A) have been reported [379,380,381,382,383,384,385].

MCAD deficiency is the most frequent disorder of FAO [386,387]. More than 500 sequence variations of MCAD have been reported so far; almost half of them are pathogenic or likely pathogenic. However, approximately 80% to 90% of the disease-causing sequence variations in caucasian patients are due to a single-base mutation: c.985A > G [388,389]. Compared to other variants, homozygosity for this mutation is associated with the most severe phenotype, including sudden infant death [386,390]. Moreover, the same single-base mutation in MCAD (c.985A > G) was observed in patients with Reye syndrome and Reye-like syndromes. However, the consequences and importance of these associations are not fully understood [391,392,393].

LCHAD deficiency (LCHADD) is diagnosed when a mutation in the alpha subunit of mitochondrial trifunctional protein (HADHA) causes an isolated deficiency of LCHAD. The most abundant pathogenic mutation is c.1528-G > C. This missense variation causes a loss of enzyme activity without changing the conformation and assembly of the MTP complex [394,395]. Although the worldwide LCHADD prevalence is estimated at 1/250,000 in Baltic Sea areas, the frequency is higher, especially in the Pomeranian district (1/20,000) [396].

Mitochondrial trifunctional protein deficiency (MTPD) is diagnosed when mutations in HADHA or HADHB (a beta subunit of mitochondrial trifunctional protein) genes lead to a deficiency of all enzyme activities in the MTP complex. According to the Orphanet database, MTPD has been reported in less than 100 cases (Orphanet). Although the clinical manifestations of pathogenic variants of HADHA and HADHB are similar, it is more likely that patients with HADHA mutations will have a severe/lethal phenotype [397]. Moreover, the survival rate for MTPD is lower than LCHADD [398,399]. In some cases, HELLP syndrome (hemolysis, elevated liver enzymes, lowered platelets) may occur in pregnant women carrying a fetus with HADHA or HADHB pathogenic mutations [400,401].

Most of the pathogenic variants of CPT1A result in undetectable or extremely low enzymatic activity [402,403]. Although CPT1 deficiency is very rare in the general population, the frequency of the milder phenotype c.1436C > T (p.P479L) is much higher in Inuit, Alaskan Native, and Canadian First Nations (even up to 1.3/1000) [404,405]. Spastic paraplegia 73 is a neurodegenerative disorder characterized by slow, gradual, and progressive weakness, and spasticity of the lower limbs is caused by mutations in CPT1C. Up to 2019, only two families were diagnosed with it. Minimal data suggested that pathogenic mutations destabilize the interaction between the regulatory and catalytic domains of the enzyme [406,407].

CPT2 deficiency has three clinical forms: lethal neonatal, severe infantile, and myopathic (which may manifest from infancy to adulthood). The myopathic form is the most common and the least severe [408,409]. In some individuals, even heterozygous pathogenic mutations may give symptoms of the myopathic form when accompanied by specific triggers (e.g., excessive exercise) [410]. Moreover, some single-base mutations in CPT2 are associated with susceptibility to infection-induced acute encephalopathy 4 [411,412].

A lack of a functional OCTN2 carnitine transporter in cell membranes leads to primary carnitine deficiency, an autosomal recessive disorder of FAO, which has a frequency of 1:40,000–1:100,000 in newborns. The absence of the cell membrane carnitine transporter causes (a) urinary carnitine wasting, (b) a significant decrease in intracellular carnitine concentration, and (c) decreased plasma-free carnitine (0–5 µmol/L in patients with primary carnitine deficiency versus 25–50 µmol/L in healthy patients) and acylated carnitine [67]. Younger children with primary carnitine deficiency display problems with (a) feeding, (b) respiratory infection, and (c) acute gastroenteritis (so-called metabolic syndrome). Later on, patients become lethargic and have hepatomegaly. Laboratory examination usually reveals (a) hypoglycemia with minimal or no KBs in urine and (b) hyperammonemia. Older patients dominate cardiomyopathy. Sometimes, older patients display both metabolic and cardiac symptoms. Moreover, a few patients with primary carnitine deficiency have been completely asymptotic for all of their lives. Primary carnitine deficiency can be successfully treated by carnitine supplementation (usually 100–400 mg per kg body weight per day) if the treatment is started before organ damage occurs. Unfortunately, a high dose of carnitine has side effects, like diarrhea and intestinal discomfort [67].

In SCAD deficiency (SCADD), mild, moderate, and severely decreased enzyme function can be observed despite no correlation between the clinical phenotype and the degree of SCAD dysfunction. The most common variations in SCADD patients, c.511C > T and c.625G > A, are also present in approximately 14% of the general population. The rarity of ACADS inactivating variants and the lack of clinical significance in many patients lead to questions regarding the clinical relevance of SCADD as a hereditary disease [386,413,414,415].

Upon closer examination, all patients diagnosed with LCAD dehydrogenase deficiency before 1992 were shown to have a defect in VLCAD [416,417,418]. More than 90 pathogenic variations in VLCAD were identified, with the c.848T > C pathogenic variant as the most frequent [419]. Sequence variations associated with a complete loss of function result in death in the first few days of life [386].

The prevalence of mitochondrial short-chain enoyl-CoA hydratase 1 deficiency (ECHS1D) remains unknown since it is a sporadic disease with less than 50 cases worldwide (data until 2020) [420,421]. Pathogenic variants of ECHS1 lead to a decrease in enzyme activity. The degree of function loss can vary and determines the severity of clinical symptoms [422,423]. In some patients with Leigh syndrome, a severe neurological disorder was caused by mutations in succinate dehydrogenase complex and/or genes related to the oxidative phosphorylation pathway, and sequence variations in ECHS1 were also observed [424,425,426]. Moreover, ECHS1D has also been described in rare cases of patients with severe neonatal lactic acidosis, cardiomyopathy, cutis laxa, and exercise-induced dystonia [427,428,429,430].

Fanconi renotubular syndrome is a family of related diseases characterized by the dysfunction of proximal tubular epithelial cells, leading to the urinary leak of essential metabolites, and the different syndrome types indicate in which gene the mutation occurred. In Fanconi renotubular syndrome type 3, a single base substitution in enoyl-CoA hydratase and 3-hydroxyacyl CoA dehydrogenase (EHHADH) leads to a missense mutation. Mutated EHHADH can localize mainly in mitochondria rather than peroxisomes and functionally disrupt the MTP complex [431,432]. EHHADH deficiency may also lead to clinical symptoms resembling Zellweger syndrome, a rare peroxisome biogenesis disorder [433,434].

As a consequence of a better understanding of the biochemical traits (especially activity toward substrates with different chain lengths) of mitochondrial types of HAD, many patients initially diagnosed with SCHAD deficiency based on their symptoms were suffering from HAD deficiency (HADD) [386,435]. Pathogenic mutations in the coding sequence, introns, or regulatory regions severely reduce HAD activity, mainly in the liver [436,437,438]. Mutations in HAD are observed in less than 1% of all familial hyperinsulinemia hypoglycemia cases [439,440,441].

Peroxisomal acyl-CoA oxidase deficiency occurs due to the defects in ACOX1. As a consequence of clinical and biochemical features resembling neonatal adrenoleukodystrophy, this disorder is also known as pseudoneonatal adrenoleukodystrophy (pseuso-NALD). Until 2022, only around 30 patients with pseudo-NALD were reported in the literature [442]. Pseudo-NALD causes increased levels of VLCFAs in the tissues and plasma of the patients, while BCFAs remain at normal levels [443].

X-ALD is an X-linked inherited disease associated with severe morbidity and mortality in most affected subjects. It is characterized by impaired peroxisomal βOX of VLCFAs (C22 and more), which is reduced to approx. 30% of healthy subjects [444]. It is a disease with a frequency in 1:17,000 newborns and is caused by mutations in the *ABCD1* gene located on the X-chromosome [445,446]. Mutations in the *ABCD1* gene (approx. 600 different mutations have been identified so far) cause the absence or dysfunction of this transporter.

Consequently, the accumulation of VLCFAs in plasma and tissues/organs, including the brain’s white matter, the spinal cord, and the adrenal cortex, occurs. Accumulated VLCFAs in tissue/organs are toxic because they disrupt cell membranes’ structure, stability, and function. So far, there is no treatment for most patients with X-ALD [447]. However, studies conducted on Abcd1 knock-out mice and human and mouse X-ALD fibroblasts revealed that overexpression of *abcd2* or *abcd3* may restore peroxisomal VLCFA β-oxidation [448].

## 5. Conclusions

The data presented in this review indicate the importance of βOX in an increasing number of tissues and organs, even those previously not considered important. Disturbances in βOX and the α- and ω-oxidation of FAs, including those caused by genetic defects, play an important role in developing various diseases. Several studies also indicate that carcinogenesis is associated with significant disturbances in βOX. Thus, deeper knowledge of the mechanisms linking a disturbance in βOX to several pathologies, including carcinogenesis, is needed to identify novel diagnostic markers and potential therapeutic interventions that may optimize the clinical management of patients with βOX and the α- and ω-oxidation of FA-related disorders.

## Figures and Tables

**Figure 2 ijms-24-14857-f002:**
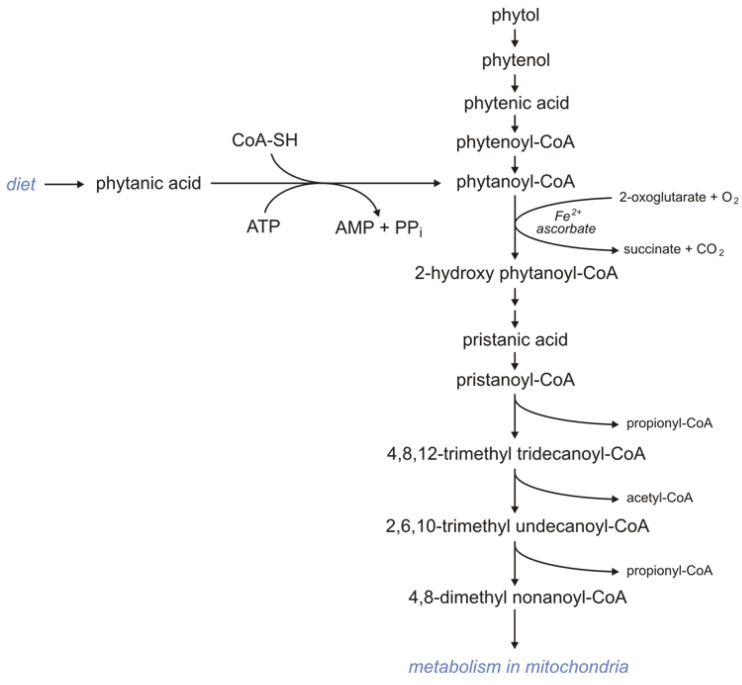
Metabolism of phytanic acid.

**Figure 3 ijms-24-14857-f003:**
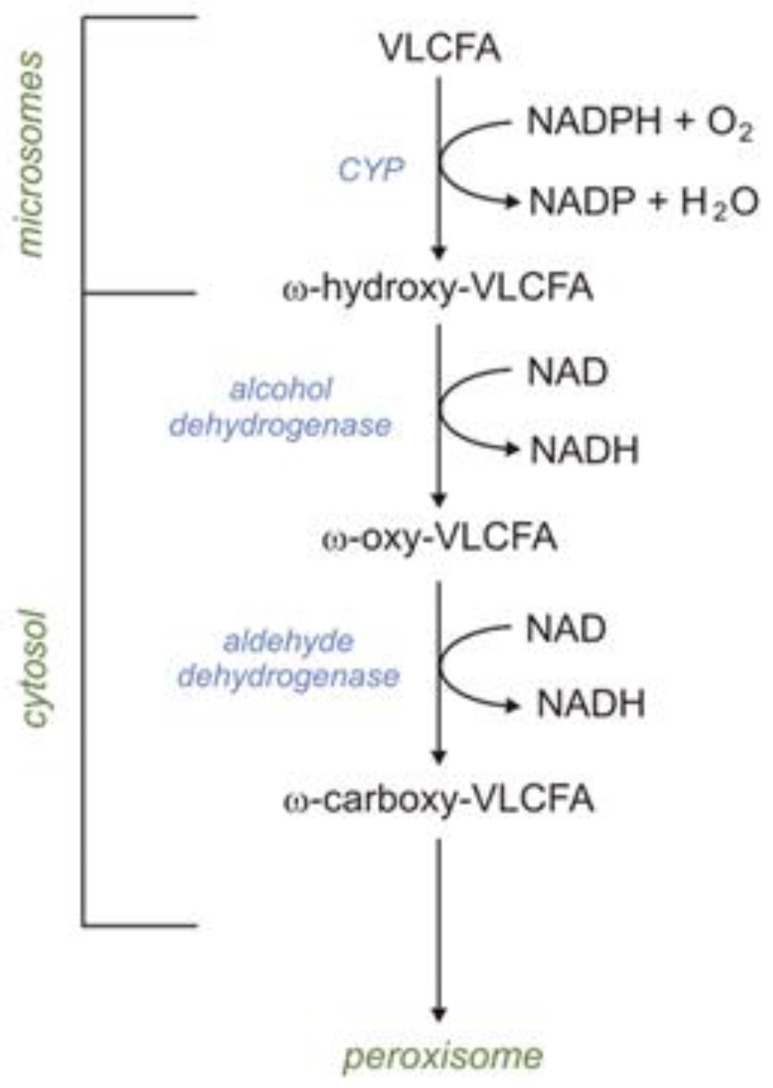
Omega oxidation of very-long-chain fatty acids (VLCFAs).

**Figure 4 ijms-24-14857-f004:**
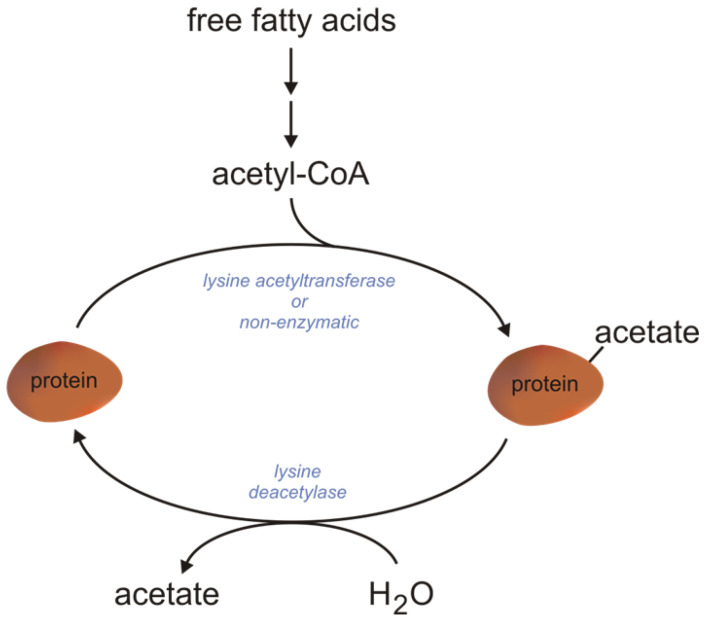
The role of free fatty acids in the acetylation of proteins.

**Figure 5 ijms-24-14857-f005:**
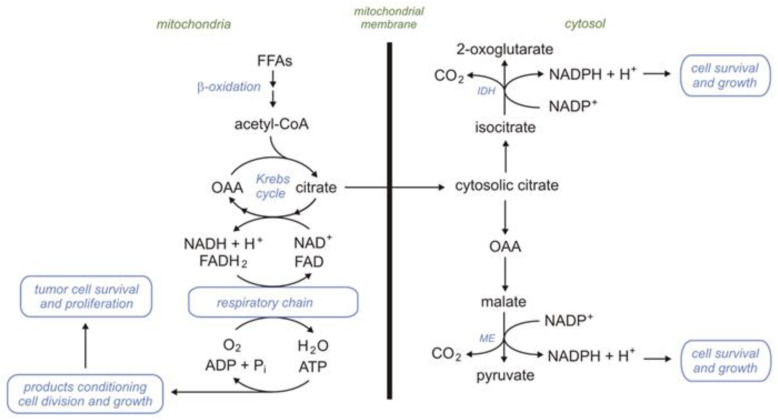
The role of FAO in cancer cell survival and growth. FFAs—free fatty acids, OAA—oxaloacetic acid, IDH—isocitrate dehydrogenase, ME—malic enzyme.

**Table 1 ijms-24-14857-t001:** Characteristics of acyl-CoA synthetases. ACSL—long-chain acyl-CoA synthetase, ACSM—medium-chain acyl-CoA synthetases, ACSS—short-chain acyl-CoA synthetases, ACSVL—very-long-chain acyl-CoA synthetase, BAT—brown adipose tissue, ER—endoplasmic reticulum, WAT—white adipose tissue.

Name/Abbreviation	Organ/Tissue Localization	Subcellular Compartment	References
ACSVL [FATP2]	Liver, intestine, kidneys, brain	Peroxisomes, ER	[34]
ACSVL [FATP6]	Heart	Cytosol, plasma membrane	[35]
ACSVL [FATP3]	Lungs, gonads, adrenals	ER, mitochondrial membrane	[34]
ACSVL [FATP1]	Skeletal muscles, BAT, WAT, heart	Plasma membrane	[36]
ACSVL [FATP4]	Skeletal muscles, BAT, WAT, intestine, skin	Peroxisomes, ER, mitochondrial membrane	[37]
ACSVL [FATP5]	Liver	Plasma membrane	[38]
ACSL1	Liver, heart, BAT, WAT, skeletal muscles	Mitochondria (outer mitochondrial membrane on the cytosolic side), lipid droplets, microsomes, plasma membrane	[39]
ACSL3	Brain, gonads, small amounts in other tissues (liver)	Lipid droplets, the cytoplasmic face of ER, the outer mitochondrial membrane	[40]
ACSL4	Adrenals, ovaries, testes, liver, skeletal muscles, small amounts in the brain	Endosomes, peroxisomes, plasma membrane, secretory vesicles, ER regions in close contact with mitochondria—mitochondrial-associated membranes	[41]
ACSL5	BAT, the duodenal mucosa, liver, skeletal muscles, kidneys, lungs	The outer mitochondrial membrane on the cytosolic side	[42]
ACSL6	Ovaries, testes, brain, skeletal muscles, small amounts in the WAT, kidneys, the duodenal mucosa	Plasma membrane	[43]
ACSM	Liver, skeletal muscles, cardiomyocytes, colonocytes, kidneys	Mitochondria. All ACSMs belong to a group of enzymes called XM-ligases (xenobiotic/medium-chain fatty acid-CoA ligases)	[44,45]
ACSS1	Brain, blood, testes, intestine, heart, kidneys, skeletal muscles, BAT	Mitochondria. ACSS1 activates acetate	[46]
ACSS2	Liver and kidneys	Cytosol, nucleus. ACSS2 activates acetate. ACSS2 is downregulated during fasting	[46,47]
ACSS3	Liver	Mitochondria. ACSS3 has a higher affinity for propionate. ACSS3 is upregulated in the fasting state	[30,46]

**Table 2 ijms-24-14857-t002:** Characteristics of acyl-CoA dehydrogenases. ACAD9—acyl-CoA dehydrogenase DH-9, BCFA—branched-chain fatty acid, LCAD—long-chain acyl-CoA dehydrogenase, LCFA—long-chain fatty acid, MCAD—medium-chain acyl-CoA dehydrogenase, MCFA—medium-chain fatty acid, SCAD—short-chain acyl-CoA dehydrogenase, SCFA—short-chain fatty acid, VLCAD—very-long-chain acyl-CoA dehydrogenase, VLCFA—very-long-chain fatty acid.

Enzyme	Mitochondrial Compartment	Preferred Substrates (Acyl-CoAs)	Tissue/Organ/Cell	Reference
VLCAD	Inner mitochondrial membrane	LCFA (mainly palmitoyl-CoA) and VLCFA (C14–C22)	Muscles, heart, liver, skin fibroblasts	[84]
Acyl-CoA DH-9 (ACAD9)	Inner mitochondrial membrane	Unsaturated LCFA, VLCFA (C16:1, C18:1, C18:2; C22:6)	Brain, liver, heart, skeletal muscle	[85]
LCAD	Matrix	LCFA, unsaturated MCFA, SCFA, BCFA (in vitro)	Lungs—pulmonary surfactant	[86]
MCAD	Matrix	MCFA (C6:0–C12:0)	Heart, skeletal muscles, liver	[87]
SCAD	Matrix	SCFA (mainly butyryl-CoA); MCFA (C6:0–C12:0)	Liver, heart, skeletal muscle	[88]

**Table 3 ijms-24-14857-t003:** Comparison between peroxisomal and mitochondrial β-oxidation. ABCD1–4—ATP-binding cassette sub-family D 1–4, ACADs—acyl-CoA dehydrogenases, ACOXs—acyl-CoA oxidases, BCFA—branched-chain fatty acid, CPT1—carnitine palmitoyltransferase 1, CPT2—carnitine palmitoyltransferase 2, CAC—acylcarnitine translocase, FAs—fatty acids, VLCADs—very-long-chain fatty acids, LCFAs—long-chain fatty acids, MCFAs—medium-chain fatty acids, PUFAs—polyunsaturated fatty acids, SCFAs—short-chain fatty acids, H_2_O—hydrogen peroxide, ETF—electron-transferring flavoprotein, OXPHOS—oxidative phosphorylation.

	Peroxisomal β-Oxidation	Mitochondrial β-Oxidation	References
Proteins involved in the transport of FAs to peroxisomes/mitochondria	ABCD1, ABCD2, and ABCD3	Carnitine transport system (CPT1, CPT2, CAC)	[115,116]
Substrates	VLCFAs (>C22), BCFAs (e.g., pristanic acid), PUFA, 2-hydroxy FAs, long-chain dicarboxylic acids, bile acid intermediates, and a number of prostanoids	VLCFAs (≤22), LCFAs, MCFAs, and SCFAs	[117,118]
Enzyme catalyzing the first reaction	ACOXs The transfer of electrons from FADH_2_ to oxygen results in the production of H_2_O_2_, which is subsequently cleaved by peroxisomal catalase	ACADs The electrons that originate from FADH_2_ are transported to ETF, the ETF dehydrogenase, and transferred to OXPHOS. Finally, they reduce oxygen to water, which results in the production of energy in the form of ATP	[82,110]
β-oxidation end products	Acetyl-CoA, NADH, MCFAs, and FADH_2_	Acetyl-CoA, NADH, and FADH_2_	[94,110]

**Table 4 ijms-24-14857-t004:** Changes in FAO enzymes and fatty acid-binding protein gene expression in various cancers. ACAD9—acyl-CoA dehydrogenase DH-9, ACSL4—long-chain acyl CoA synthetase 4, AR—androgen receptor, CPT—carnitine palmitoyl transferase, ECH—enoyl-CoA-hydratase, EHHADH—enoyl-CoA hydratase and 3-hydroxyacyl CoA dehydrogenase, ESR—estrogen receptor, FABP—fatty acid-binding protein, HADH—3-hydroxyacyl-CoA dehydrogenase, HADHA—hydroxyacyl-CoA dehydrogenase/3-ketoacyl-CoA thiolase/enoyl-CoA hydratase (trifunctional protein), alpha subunit, LCAD—long-chain acyl-CoA dehydrogenase, SCAD—short-chain acyl-CoA dehydrogenase.

Gene/Enzyme	Nature of Change	Type of Evaluation	Cancer Type	References
ACAD9	Upregulated	mRNA level	Glioblastoma multiforme	[289]
ACSL1	Downregulated	mRNA level	Lung cancer, breast cancer	[290,291]
	Upregulated	mRNA level	Rectal adenocarcinoma, colon cancer, hepatocellular carcinoma	[290,292,293,294]
ACSL3	Downregulated	mRNA level	Ovarian cancer	[290]
	Upregulated	mRNA level	Melanoma, ESR-negative breast cancer	[290,295]
		Protein level	Large-cell lung cancer, small-cell lung cancer	[296]
ACSL4	Downregulated	mRNA and protein levels	Gastric cancer	[297]
		mRNA level	Lung cancer	[290]
	Upregulated	mRNA level	Colorectal cancer, ESR-negative breast cancer, triple-negative breast cancer, AR-negative prostate, hepatocellular carcinoma	[290,292,298,299,300,301]
		Protein level	Prostate cancer	[302]
		mRNA and protein levels	Colon adenocarcinoma, hepatocellular carcinoma	[303,304]
ACSL5	Downregulated	mRNA level	Breast cancer	[290]
		mRNA and protein levels	Small intestine cancer	[305]
	Upregulated	mRNA level	Bladder cancer, colorectal cancer	[290,306,307]
ACSL6	Downregulated	mRNA level	Leukemia	[290]
	Upregulated	mRNA level	Colorectal cancer	[290,308]
CPT1A	Upregulated	Protein level	Gastric cancer	[309]
		mRNA level	Glioblastoma multiforme	[289]
CPT1B	Upregulated	mRNA and protein levels	Prostate cancer	[310]
		mRNA level	High-grade bladder cancer	[311]
CPT1C	Upregulated	mRNA level	Gastric cancer, lung cancer, papillary thyroid carcinoma	[312,313,314]
CPT2	Downregulated	mRNA level	Hepatocellular carcinoma, colorectal cancer, ovarian cancer	[308,315,316]
	Upregulated	mRNA level	Glioblastoma multiforme	[289]
ECH1	Downregulated	mRNA level	Colorectal cancer	[317]
EHHADH	Downregulated	mRNA and protein levels	Hepatocellular carcinoma	[318]
	Upregulated	mRNA level	Osteosarcoma	[319]
FABP3	Upregulated	mRNA and protein levels	Non-small-cell lung cancer	[320]
FABP4	Downregulated	mRNA level	Stomach adenocarcinoma	[321]
		mRNA and protein levels	Hepatocellular carcinoma	[322]
	Upregulated	Protein level	High-grade serous ovarian carcinoma, pancreatic ductal adenocarcinoma, gastric adenocarcinoma	[323,324,325]
		mRNA and protein levels	Non-small-cell lung cancer, prostate cancer	[320,326]
FABP5	Upregulated	Protein level	Gastric adenocarcinoma	[325]
HADH	Downregulated	Protein level	Gastric cancer	[327]
		mRNA level	Gastric cancer, kidney renal clear cell carcinoma	[328,329,330]
	Upregulated	mRNA level	Colon cancer, acute myeloid leukemia	[331,332]
HADHA	Downregulated	mRNA level	Breast cancer	[333]
LCAD	Downregulated	mRNA level	Hepatocellular carcinoma	[334]
MCAD	Upregulated	Protein level	Glioblastoma, squamous cell carcinoma of the head and neck	[335,336]
SCAD	Downregulated	mRNA level	Colorectal cancer	[317]

**Table 5 ijms-24-14857-t005:** Cancer cell FAO as a potential therapeutic target. ACSL—long-chain fatty acid synthetase, ACSVL—very-long-chain fatty acid synthetase, CPT—carnitine palmitoyltransferase, ECHS—enoyl-CoA hydratase short chain 1, MCAD—medium-chain acyl-CoA dehydrogenase, PP2—4-amino-5-(4-chlorophenyl)-7-(t-butyl)pyrazolo[3,4-d]pyrimidine.

Targeted Enzyme	Inhibitor/Interfering Compound	Experimental Models	Effects	References
ACSL4	Rosiglitazone	Breast cancer cell lines	Inhibition of cancer cell growth	[356]
	PRGL493	Breast cancer cell lines, prostate cancer cell lines	Inhibition of cancer cell growth and sensitization to chemotherapy	[357]
ACSL5	Triacsin C	Glioma cell lines	Inhibition of cancer cell survival	[358]
	Small interfering RNA	Lung cancer cell lines	Inhibition of cancer cell growth	[359]
ACSVL3	Small interfering RNA	Glioblastoma cell lines	Inhibition of cancer cell growth and tumourigenicity	[360]
CPT1	Avocatin B	Primary myeloid leukemia cells	Inhibition of cancer cell survival	[361]
	Etomoxir	Leukemia, breast, prostate, colorectal cancer cell lines, and the xenograft model	Inhibition of cancer cell growth, survival, and tumourigenicity	[288,362,363,364,365]
		Lung cell lines	Sensitization to radiation	[366]
	Oxfenicine	Melanoma cell lines	Inhibition of cancer cell growth	[367]
	Small interfering RNAs	Brest cancer cell lines	Inhibition of cancer cell survival	[368]
CPT2	Aminocarnitine	Glioma cell lines	Inhibition of cancer cell growth	[369]
	Perhexiline	Gastrointestinal cancer cell lines	Inhibition of cancer cell survival and sensitization to chemotherapy	[370]
ECHS1	Small interfering RNA, PP2	Breast cancer cell lines	Inhibition of cancer cell survival	[371]
MCAD	Hairpin RNA interference	Glioblastoma cell lines	Inhibition of cancer cell survival	[335]

## Data Availability

No new data were created.

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
