# Peer review of "The Physiological and Pathological Role of Acyl-CoA Oxidation"

_ijms, 2023, doi:10.3390/ijms241914857_

Round 1

Reviewer 1 Report

The review titled "Physiological and Pathological role of Acyl-CoA oxidation" is an intriguing synthesis work that addresses a significant issue. 

The review is well-organized, presenting information in a logical sequence. 

It is also informative and offers an excellent scientific synthesis.

Areas for improvement in the review:

The author has to add the transcription regulation of CPT according to the level and types of fatty acids.

The author needs to include the signaling pathway of PPARalpha and explain more this part

The author has to add a table comparison between mitochondrial FAO and peroxisomal FAO.

It is also crucial to add the transporters ABCD and their regulation depending on fatty acid types.

It is necessary to add the PPARgamma and PPARbeta regulation 

 Additionally, the author has to develop Osteoclast and Pancreatic β-cell parts because they are very poor in scientific information

- Furthermore, the author should discuss the ACOX deficiency part (P-NALD)

Line 230: PGC1 is a cofactor; it isn't a transcription factor 

The author must show the major revisions made in the text by highlighting the changes in a different colored text. 

It is imperative to consider all these remarks to reinforce the manuscript's quality and conclude with more accuracy.

Author Response

Reviewer 1

Comments and Suggestions for Authors

The review titled "Physiological and Pathological role of Acyl-CoA oxidation" is an intriguing synthesis work that addresses a significant issue.

The review is well-organized, presenting information in a logical sequence.

It is also informative and offers an excellent scientific synthesis.

 Areas for improvement in the review:

  1. The author has to add the transcription regulation of CPT according to the level and types of fatty acids.

Re: This has been added on  page 6, lines 208-214

  1. The author needs to include the signaling pathway of PPARalpha and explain more this part

Re: More information about PPAR signalling has been added (page 2, lines 65-90)

  1. The author has to add a table comparison between mitochondrial FAO and peroxisomal FAO.

Re: This table has been added (table 3 in revised manuscript).

  1. It is also crucial to add the transporters ABCD and their regulation depending on fatty acid types.

Re: This information has also been added to our manuscript (page 13, lines 481-496)

  1. It is necessary to add the PPARgamma and PPARbeta regulation

Re: more information about PPAR signalling has been added (page 2, lines 65-90)

  1. Additionally, the author has to develop Osteoclast and Pancreatic β-cell parts because they are very poor in scientific information

Re: The pharagraps about Osteoclast and Pancreatic β-cell has been extended – see pages 23-24

  1. Furthermore, the author should discuss the ACOX deficiency part (P-NALD)

Re: this part of the text has been removed to page 31, and discussed more thoroughly.

                Line 230: PGC1 is a cofactor; it isn't a transcription factor

Re: PGC1 is a transcriptional coactivator that regulates the genes involved in energy metabolism. It has been corrected in revised manuscript (line 261)

The author must show the major revisions made in the text by highlighting the changes in a different colored text.

Re: The revisions are shown in track changes mode

It is imperative to consider all these remarks to reinforce the manuscript's quality and conclude with more accuracy.

Re: Thank you very much for these constructive comments. We hope that they significantly improved our Review paper.

Reviewer 2 Report

General comments:

The introduction section is very long (nearly half the manuscript) and contains to a relatively large extent common textbook knowledge. The authors cite other reviews; or primary literature, but they do often not discuss details of results from recent primary literature.

For example:

Section 1.3.2. describe the additional enzymes required for oxidation of unsaturated acyl-CoA. The 2 isoforms of the enoyl-CoA isomerase (ECI) are mentioned (the only information in this section that is not common textbook knowledge), but without given the reference immediately, but only at the end of the section. At the end of the section the authors cite the references [82 to 84]. However, the authors do not at all discuss the results from the cited references: reference [83] describe phenotype of ECI1 knock out mice with indications for a possible functional redundancy of enoyl-CoA isomerases. Ref. [84] describes the structure of human ECI; however the authors of the review do not discuss results of that paper, nor did they mention at all that its describes crystal structure of the enzyme.

Minor comments:

Figure 6: The two short lines that indicate the mitochondrial membranes are somewhat misleading (at first sight it one could think that they are crossing out the arrow of the citrate transport). Explain ME (malate enzyme) in the Figure legend.

lines 631ff: I think "II phase" should be "phase II" ?

lines 1093/1094: This sentence probably refers to Table 3 not Table 4 ?

Author Response

Reviewer 2

General comments:

  1. The introduction section is very long (nearly half the manuscript) and contains to a relatively large extent common textbook knowledge.

Re: We have removed Figure 1 and some fragments of text from the introduction to avoid quoting the textbook information. However, since this review paper is intended for a wide range of readers, we believe that recalling some fundamental information will make it easier to read.

  1. The authors cite other reviews; or primary literature, but they do often not discuss details of results from recent primary literature.

Re: The article is extensive, and now it contains more than 400 references – that is why sometimes the previous review articles which focused on some topics are cited instead of all original articles related to the problem.

For example:

Section 1.3.2. describe the additional enzymes required for oxidation of unsaturated acyl-CoA. The 2 isoforms of the enoyl-CoA isomerase (ECI) are mentioned (the only information in this section that is not common textbook knowledge), but without given the reference immediately, but only at the end of the section. At the end of the section the authors cite the references [82 to 84]. However, the authors do not at all discuss the results from the cited references: reference [83] describe phenotype of ECI1 knock out mice with indications for a possible functional redundancy of enoyl-CoA isomerases. Ref. [84] describes the structure of human ECI; however the authors of the review do not discuss results of that paper, nor did they mention at all that its describes crystal structure of the enzyme.

Re: This fragment has been rewritten and the missing information has been added (lines 340-342).

Minor comments:

  1. Figure 6: The two short lines that indicate the mitochondrial membranes are somewhat misleading (at first sight it one could think that they are crossing out the arrow of the citrate transport). Explain ME (malate enzyme) in the Figure legend.

Re: the figure (Fig. 5 in current version) has been modified according to Reviewer suggestion. ME has been explained in figure legend.

  1. lines 631ff: I think "II phase" should be "phase II" ?

Re: This has been corrected in the whole manuscript.

  1. lines 1093/1094: This sentence probably refers to Table 3 not Table 4 ?

Re: As we have added one more table in revised manuscript (table 3) the reference to table 4 is now correct.

Reviewer 3 Report

In their present review, the authors describe the importance of fatty acid metabolism for physiology and pathology in different tissues and in different diseases. The mechanistic differences between the alpha, beta and omega oxidation of FFA in the peroxisome, the mitochondrion and the ER as well as tissue-specific differences are discussed. In addition, an overview of FFA uptake in different tissues and organelles is given. A focus of this review is the FAO in different cancers and the importance of individual genes involved in FFA metabolism. The review offers a comprehensive but not too detailed overview of the topics covered and is a good introduction to the different aspects of the FAO. The diagrams and tables are clear and helpful for understanding. The text is generally well written and the selected literature represents a good selection of the literature published on the topics. Since the FAO in pancreatic beta cells is at least very briefly mentioned (paragraph 2.6.8), I would like to draw attention to an interesting aspect that should be mentioned in the present review. The FAO of LCFA and VLCFA in the peroxisomes of pancreatic beta cells is involved in the cell damage and apoptosis of these cells and take part in the manifestation of T2DM. T2DM is very often associated with metabolic syndrome, which is characterized, among other things, by increased plasma concentrations of long-chain free fatty acids. These are increasingly metabolized in the peroxisomes, which, as clearly described in this manuscript, leads to an increased formation of hydrogen peroxide. However, this cannot be sufficiently detoxified in pancreatic beta cells, as almost no expression of catalase can be detected in these cells.

Lit

Peroxisome-generated hydrogen peroxide as important mediator of lipotoxicity in insulin-producing cells.

Elsner M, Gehrmann W, Lenzen S. Diabetes. 2011 Jan;60(1):200-8. doi: 10.2337/db09-1401. Epub 2010 Oct 22.

Oxidative stress: the vulnerable beta-cell

Sigurd Lenzen

DOI: 10.1042/BST0360343

A list of abbreviations would be helpful for easier readability.

Author Response

Reviewer 3

In their present review, the authors describe the importance of fatty acid metabolism for physiology and pathology in different tissues and in different diseases. The mechanistic differences between the alpha, beta and omega oxidation of FFA in the peroxisome, the mitochondrion and the ER as well as tissue-specific differences are discussed. In addition, an overview of FFA uptake in different tissues and organelles is given. A focus of this review is the FAO in different cancers and the importance of individual genes involved in FFA metabolism. The review offers a comprehensive but not too detailed overview of the topics covered and is a good introduction to the different aspects of the FAO. The diagrams and tables are clear and helpful for understanding. The text is generally well written and the selected literature represents a good selection of the literature published on the topics. Since the FAO in pancreatic beta cells is at least very briefly mentioned (paragraph 2.6.8), I would like to draw attention to an interesting aspect that should be mentioned in the present review. The FAO of LCFA and VLCFA in the peroxisomes of pancreatic beta cells is involved in the cell damage and apoptosis of these cells and take part in the manifestation of T2DM. T2DM is very often associated with metabolic syndrome, which is characterized, among other things, by increased plasma concentrations of long-chain free fatty acids. These are increasingly metabolized in the peroxisomes, which, as clearly described in this manuscript, leads to an increased formation of hydrogen peroxide. However, this cannot be sufficiently detoxified in pancreatic beta cells, as almost no expression of catalase can be detected in these cells.

Re: The aspect raised by the Reviewer has been added to the revised manuscript (lines 1068-1078).

A list of abbreviations would be helpful for easier readability.

Re: As the list of abbreviations is quite long it has been added to supplementary materials.

Round 2

Reviewer 1 Report

In this version of the review, “Physiological and Pathological role of Acyl-CoA oxidation.” We can see an acceptable evolution compared to the first version because it has become more structured with more explanation.

the authors have taken the reviewer's remarks and suggestions into consideration, which has positively impacted the quality and consistency of the article.

with this version, the article shows an excellent scientific level and represents an added value in the interested research topics 

the article is accepted for me with this version